# Effects of Local Mixing Ratios and Mass Flow Rates on Combustion Performance of the Fuel-Rich LOX (Liquid Oxygen)/Kerosene Gas Generator in the ATR (Air Turbo Rocket) Engine

Yuankun Zhang [1,2], Qingjun Zhao [1,2,3], Bin Hu [1,2,*], Qiang Shi [1], Wei Zhao [1,2] and Xiaorong Xiang [1,2]

1   Institute of Engineering Thermophysics, Chinese Academy of Sciences, Beijing 100190, China
2   School of Aeronautics and Astronautics, University of Chinese Academy of Sciences, Beijing 100190, China
3   Beijing Key Laboratory of Distributed Combined Cooling Heating and Power System, Beijing 100190, China
*   Correspondence: hubin@iet.cn

**Abstract:** This paper presents a numerical simulation analysis of the flow and combustion characteristics of a fuel-rich LOX (liquid oxygen)/kerosene gas generator in an ATR (air turbo rocket) engine, examining the effects of local parameters on the combustion flow field and performance. The analysis considers variations in unit injector mixing ratios and unit mass flow rates. The results indicate that as the mixing ratio in the inner-ring injectors increases (while the mixing ratio in the middle-ring injectors decreases), the oxygen concentration area near the axis zone and the 50% radius zone of the gas generator expands. Conversely, the kerosene concentration area near the axis zone decreases while gradually increasing near the 50% radius zone. In the flow direction section, there is an inverse relationship between the variation trend of local temperature and the oxygen concentration in the local area. As the oxygen concentration increases, the temperature decreases. The temperature distribution across the cross-section of the gas generator exhibits a circular pattern. When the mixing ratio (or mass flow rates) of the unit injector are perfectly balanced, the temperature distribution becomes highly uniform. A larger disparity in flow rate between the inner ring injector and the middle ring injector leads to a lower combustion efficiency. This effect differs from the effect of the mixing ratio difference between the two injector rings. Increasing the mixing ratio in the inner-ring injectors (or decreasing the mixing ratio in the middle-ring injectors) initially leads to a rise in combustion efficiency, followed by a subsequent decline. The maximum combustion efficiency of 89.10% is achieved when the mixing ratio is set to $Km_{-1} = 0.7$ and $Km_{-2} = 2.79$, respectively. Increasing the flow rate in the inner-ring injectors (or decreasing the flow rate in the middle-ring injectors) initially leads to an improvement in combustion efficiency, followed by a subsequent reduction. The maximum combustion efficiency of 86.13% is achieved when the mass flow rate is set to $m_{-1} = m_{-2} = 0.1$ kg/s.

**Keywords:** ATR engine; fuel-rich gas generator; bipropellant injector; unit injector mixing ratio; unit injector mass flow rate; combustion efficiency

## 1. Introduction

The air turbo rocket (ATR) engine is an air-breathing, wide-range combined propulsion device that consists of an inlet, compressor, gas generator, turbine, afterburner, and nozzle [1,2]. The working principle of the ATR engine is illustrated in Figure 1. In the gas generator, LOX and kerosene undergo controlled combustion reactions to generate high-temperature and high-pressure fuel-rich combustion products. These products drive the turbine, while the turbine powers the compressor responsible for compressing the air. The compressed air passes through the outer passage, while the fuel-rich combustion products

flowing from the turbine pass through the inner passage, resulting in secondary combustion in the afterburner. The afterburner subsequently generates high-temperature and high-pressure gases, which are then accelerated through the nozzle to produce thrust [3,4]. The ATR engine combines the technical advantages of a rocket engine and an aero engine, resulting in a high thrust-to-weight ratio and a high specific impulse. As a result, the ATR engine can operate over a wide range of velocities and in a broad range of airspace [5,6]. Additionally, the compressor's air intake requirements vary under different operating conditions, and the turbine powers the compressor through the gas generator. Consequently, the gas generator in the ATR engine must possess a significant flow regulation capability to accommodate the ATR engine's operation across a wide range of conditions.

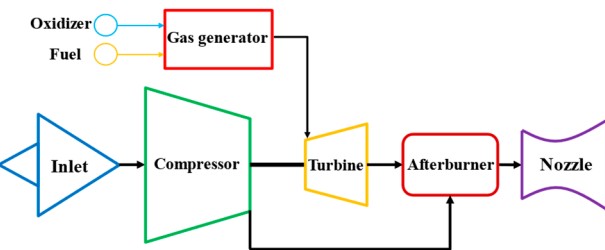

**Figure 1.** Schematic of the ATR engine.

The fuel-rich gas generator serves as the core component of the ATR engine and plays a crucial role in ensuring reliable operation and enhancing the engine's performance [7]. Consequently, extensive research has been conducted on the combustion dynamics of the fuel-rich gas generator. Wang et al. [8] used the established ATR engine working fluid calculation model to calculate the physical parameters and components of the primary gas produced by the combustion of four propellants at a given pressure and temperature, then compared and analyzed them according to the propellant selection requirements, and finally chose the appropriate propellant for the ATR engine. Wang et al. [9] numerically simulated the turbulent mixing and combustion processes of rich fuel gas and air and analyzed the influence of the air intake mode and intake parameters on the engine's performance. The results from this study revealed that both sides of air intake was better than only air intake from one side alone. The air incidence angle was 90°, and the engine performance was deemed to be ideal; the air intake position was selected between the length-to-diameter ratio of the afterburner from 1.0 to 1.5, and the combustion efficiency of the afterburner was found to have increased. Fernández et al. [10] constructed a comprehensive numerical model to examine the performance of the air turbo rocket during the supersonic acceleration of the vehicle. The performance was found to be optimal when the demand from the air turbo rocket matched the intake air flow capture. The heat recovery across the regenerator was found to be critical for the operation of the turbomachinery at a low speed. The transition of the air turbo rocket to the ramjet operation was identified at Mach 4.5. During this regime, the propulsion plant was rather insensitive to the mixture ratio and was throttled with the air turbo rocket throat area. Nan et al. [11] investigated the effects of key parameters on the thermodynamic process of ATR engines. These parameters were as follows: the temperature ratio in the gas generator, the pressure ratio in the compressor, and the pressure ratio in the turbine. On this basis, the overall performance of the ATR engines with the monopropellants and bipropellants was studied through numerical simulations and experiments. From analyzing the start-up and operation characteristics under a high-attitude working condition, it was found that a large thrust and short start-up time (of less than five seconds) can be obtained in flight envelopes with a Mach number of less than four. Lausten et al. [12] studied the formation and deposition of soot in the fuel-rich combustion process of LOX/kerosene using the experiments under the pressure range of 5–10 MPa and the mixing ratio range of 0.25–1.0, respectively. The results showed that soot deposition was not obvious when the mixing ratio was 0.34 and 0.25, and the soot deposition rate increased significantly when the mixing ratio was 0.57. When the mixing ratio remained at 0.35,

the change in combustor pressure had no obvious effect on the soot deposition rate. Liu et al. [13], established operation characteristic models of an air inlet and a compressor, and verified them through numerical simulations to gain the matched operation characteristics of the air inlet and the compressor in an ATR engine. Khan and Qamar et al. [14] conducted an experimental investigation of the gas generator of the liquid propellant rocket engine to determine the optimum characteristic length of the gas generator with liquid oxygen and kerosene. The experimental results of these hot tests revealed that the highest characteristic velocity of the gas generator was achieved at a characteristic length of 4.27 m. Meanwhile, the effects of the mixing ratio on the chamber pressure were discussed, with the results showing that characteristic velocity and pressure increased with increases in the oxidizer-to-fuel mixing ratio. J Yu and C Lee et al. [15] developed the calculation model for the non-equilibrium chemical reaction of kerosene/LOX in the fuel-rich gas generator, with detailed kerosene kinetics developed by Dagaut and soot formation mechanisms. These results could provide exceptionally reliable and accurate numbers in the prediction of the combustion gas temperature, species fraction, and material properties except for the CO and $H_2$ mole fractions. Based on the above conclusions, J Yu and C Lee et al. [16] then studied the turbulence generation and interactions due to the turbulence ring and splash plate in the gas generator. Their calculation results showed that the installation of turbulence ring can introduce additional turbulences and improve turbulent mixing in the downstream flow. Seo S et al. [17,18] studied the dynamic characteristics of the LOX/ kerosene (RP-1) fuel-rich combustion gas generator under the pressure of 4.10–7.24 MPa and the mixing ratio of 0.32, respectively. The experimental results showed that the coaxial swirl injector presented different combustion responses according to its configuration and working pressure under fuel-rich combustion, resulting in different damping characteristics of low-frequency fluctuations. Son M et al. [19] used turbopumps to supply the propellant design method for a kerosene fuel-rich gas generator of the liquid rocket engine. The minimum residence time in the chamber and the characteristic length were calculated by adding the reaction time and the vaporization time. Using the characteristic length, the design methods for the fuel-rich gas generator were established. Ahn K et al. [20,21] carried out an experimental study to investigate the combustion characteristics of liquid–liquid swirl coaxial injectors in a fuel-rich gas generator. It was found that the combustion characteristic velocity, combustion gas temperature, and combustion dynamics were seldom influenced by the increase of injector recess length in the present fuel-rich conditions. The results of the dynamic pressure data and swirl-injector dynamics suggest that the longitudinal mode combustion instability in the fuel-rich gas generator equipped with bi-swirl coaxial injectors could be significantly affected by the relationship between the resonant frequency in the combustion chamber and the flow dynamics of the swirl injector. Rao MR et al. [22] investigated the performance and flow field parameters of a dump combustor through conducting experiments and using an analytical model. The developed analytical model and extensive experimental data showed that sustained ignition was achieved for a particular equivalence ratio range and fuel flow rates. The stages of liquid droplet breakup, evaporation, and mixing with oxidants in heated flow were planned to be added at later stages of the study. Yao and Zhou et al. [23] established the thermal performance model of the variable-thrust liquid rocket engine and analyzed the influence of the main liquid rocket engine design parameters on the performance optimization, matched the LOX/kerosene propellants system parameters under a 100–20% variable-thrust operation, and conducted an analysis on the liquid rocket engine operating characteristics under a wide range of variable-thrust operations. Han J et al. [24] conducted their experiments to investigate the propagation characteristics of the rotating detonation wave with fuel-rich gas generator, and the effects of the basic parameters on the rotating detonation KFRG. Song et al. [25] implemented a kerosene combustion model to predict the combustion behavior in the specified oxidizer-to-fuel mixing ratio range of 0.3 to 0.4. To consider the fuel-rich condition, the three-species surrogate mixture model was selected, and the droplet evaporation and two-temperature area models were incorporated into the PSR model. Dagaut et al. [26] used three-species

surrogate mixture models to provide more detailed kinetic models that better represent the combustion behavior of kerosene, but there are issues in terms of computational efficiency.

The above review reveals that current research on gas generators primarily concentrates on injector types, characteristic length, soot formation, detailed kinetic models, and optimization of one-dimensional models. However, there is a lack of research available that investigates the impact of the local mixing ratio and the local flow rate on the combustion flow field and performance. This paper aimed to conduct numerical simulations to investigate the effects of the local mixing ratio and the local mass flow rate on the combustion performance of the LOX/kerosene gas generator in the ATR engine, while maintaining a fixed total mixing ratio and total flow rate. This study provides valuable technical support for the optimization design of the LOX/kerosene fuel-rich gas generator in the ATR engine.

## 2. Numerical Method

### 2.1. Physical Model

Figure 2a shows the physical model of the fuel-rich gas generator that was used in this study. The model primarily consists of a GO$_2$ (Gas oxygen)/kerosene torch igniter, oxidant inlet valve, fuel inlet valve, injection panel, gas generator body, and gas generator nozzle. The gas generator had a diameter of 90 mm and a length of 270 mm, while the nozzle throat diameter was 35 mm, and the characteristic length was 2.046 m, respectively. The gas generator utilizes LOX/kerosene as the propellant, while the torch igniter uses GO$_2$/kerosene.

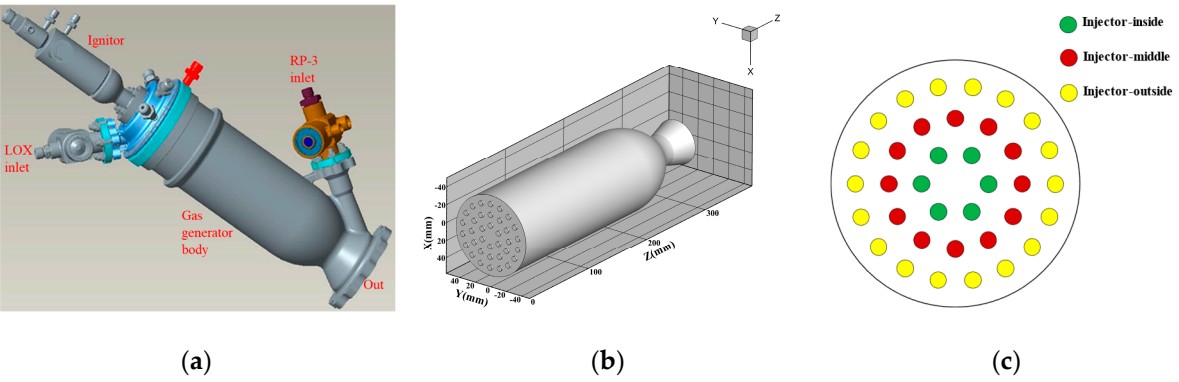

        (**a**)                             (**b**)                               (**c**)

**Figure 2.** Structural sketch of the gas generator. (**a**) Gas generator; (**b**) the computational domain of the gas generator; and (**c**) the injector arrangement of the gas generator.

Figure 2b shows the schematic diagram of the computational domain for the gas generator, which primarily consists of the injection surface, gas generator body, and nozzle. The flow direction of LOX and kerosene is along the Z-axis. Figure 2c shows the arrangement of the injectors in the gas generator. The propellant injection utilized a multi-unit swirl injector configuration, divided into three concentric circles: inner-ring (green), middle-ring (red), and outer-ring (yellow). The layout is depicted in Figure 2c. The inner and middle rings of the injectors are internal-mixing bi-propellant swirl injectors, facilitating the complete combustion of LOX and kerosene. The outer-ring injectors, on the other hand, are monopropellant (kerosene) swirl injectors that form an internal cooling film near the combustor wall. The LOX and kerosene were ignited by the torch igniter situated at the center of the propellant injection panel. The inner, middle, and outer rings contained 6, 12, and 18 injectors, respectively.

At the design point, the gas generator had a total mass flow rate of 2.26 kg/s and a mixing ratio of 1 between LOX and kerosene, respectively. This paper primarily investigated the effects of local mass flow rates and local mixing ratios on the combustion flow field. The effects of combustion and the distribution of cooling kerosene flow on wall temperature were not considered. To ensure the feasibility of the computational scheme used in this paper, the mass flow rate of the outer injectors was maintained at 0.46 kg/s under different

conditions. This ensured that the wall temperature of the gas generator does not exceed the limit (with the inner wall temperature not exceeding 1200 K). This paper employed two computational schemes, as presented in Tables 1 and 2, while maintaining a constant total mass flow rate and mixing ratio. Table 1 presents the variation of the mass flow rate for the unit injector while keeping the injector's mixing ratio fixed. In Table 2, the mixing ratio of the unit injector is altered while maintaining a constant mass flow rate for the injector.

**Table 1.** Mixing ratios of the unit injector.

| Label | $Km_{-1}$ | $m_{-1}$ (kg/s) | $Km_{-2}$ | $m_{-2}$ (kg/s) |
|---|---|---|---|---|
| 1 | 0.43 | 0.10 | 3.80 | 0.10 |
| 2 | 0.70 | 0.10 | 2.79 | 0.10 |
| 3 | 1.00 | 0.10 | 2.24 | 0.10 |
| 4 | 1.69 | 0.10 | 1.69 | 0.10 |
| 5 | 2.33 | 0.10 | 1.45 | 0.10 |

**Table 2.** Mass flow rates of the unit injector.

| Label | $Km_{-1}$ | $m_{-1}$ (kg/s) | $Km_{-2}$ | $m_{-2}$ (kg/s) |
|---|---|---|---|---|
| 6 | 1.69 | 0.02 | 1.69 | 0.14 |
| 7 | 1.69 | 0.06 | 1.69 | 0.12 |
| 8 | 1.69 | 0.10 | 1.69 | 0.10 |
| 9 | 1.69 | 0.14 | 1.69 | 0.08 |
| 10 | 1.69 | 0.20 | 1.69 | 0.05 |

The mixing ratio of the unit injector located in the inner-ring and middle-ring was defined as $Km_{-1}$ and $Km_{-2}$, respectively. The mass flow rates of the unit injector located in the inner-ring and middle-ring were defined as $m_{-1}$ and $m_{-2}$, respectively. The specific parameters of all conditions utilized are shown in Tables 1 and 2.

*2.2. Turbulence Model*

The Reynolds stress term is closed by the realizable $k$–$\varepsilon$ turbulence model. The chemical reaction equilibrium model was selected to close the turbulent chemical reaction rate. To obtain a higher accuracy, the flow equations were discretized by the second-order upwind scheme, and the SIMPLE algorithm [27] was utilized for the coupling of the pressure and the flow.

The turbulence model $k$ equation and $\varepsilon$ equation are as follows:

$$\frac{\partial}{\partial t}(\rho k) + \frac{\partial}{\partial x_j}(\rho k u_j) = \frac{\partial}{\partial x_j}\left[\left(\mu + \frac{\mu_t}{\zeta_\alpha}\right)\frac{\partial k}{\partial x_j}\right] + G_k + G_b - \rho\varepsilon - Y_M, \tag{1}$$

$$\frac{\partial}{\partial t}(\rho\varepsilon) + \frac{\partial}{\partial x_j}(\rho\varepsilon u_j) = \frac{\partial}{\partial x_j}\left[\left(\mu + \frac{\mu_t}{\zeta_\varepsilon}\right)\frac{\partial\varepsilon}{\partial x_j}\right] + \rho C_1 S\varepsilon - \rho C_2 \frac{\varepsilon^2}{k + \sqrt{\gamma\varepsilon}} + C_{1\varepsilon}\frac{\varepsilon}{k}C_{3\varepsilon}G_b, \tag{2}$$

where the parameter $C_1, \eta, S$, is defined as:

$$C_1 = \max\left[0.43, \frac{\eta}{\eta + 5}\right], \ \eta = S\frac{k}{\varepsilon}, \ S = \sqrt{2S_{ij}S_{ij}}. \tag{3}$$

*2.3. The Species Transport Model and the Chemical Reaction Model*

The species transport model of combustion adopts the eddy dissipation model [28], which considers the mutual diffusion process between the species and the diffusion effect

of the propellant inlet. The chemical reaction rate of the model is controlled by the large eddy mixing time scale $k$–$\varepsilon$, and the eddy dissipation reaction rate can then calculated.

Real kerosene is composed of hundreds of species, making it challenging to accurately depict the complex chemical reaction process. Numerical simulations commonly employ detailed reaction mechanisms, simplified reaction mechanisms, and one-step/two-step overall reaction mechanisms. This paper utilizes the 10-step reaction mechanism of kerosene proposed by Choi [29], which has been extensively validated in kerosene combustion reactions. The specific reaction is shown in Table 3.

**Table 3.** Mechanism of the kerosene-$O_2$ reaction system [a] (10-step).

| No. | Reaction | A | n | E |
|---|---|---|---|---|
| 1 | $C_{10}H_{20}+5O_2\Rightarrow10CO+10H_2$ | $2.00 \times 10^{16}$ | 0.0 | 52,000 |
| 2 [b] | $CO+O\Leftrightarrow CO_2+M$ | $5.30 \times 10^{13}$ | 0.0 | −4540 |
| 3 | $CO+OH\Leftrightarrow CO_2+H$ | $4.40 \times 10^{6}$ | 1.5 | −740 |
| 4 | $H_2+O_2\Leftrightarrow OH+OH$ | $1.70 \times 10^{13}$ | 0.0 | 48,000 |
| 5 | $H+O_2\Leftrightarrow OH+O$ | $2.60 \times 10^{14}$ | 0.0 | 16,800 |
| 6 | $OH+H_2\Leftrightarrow H_2O+H$ | $2.20 \times 10^{13}$ | 0.0 | 5150 |
| 7 | $O+H_2\Leftrightarrow OH+H$ | $1.80 \times 10^{10}$ | 1.0 | 8900 |
| 8 | $OH+OH\Leftrightarrow H_2O+O$ | $6.30 \times 10^{13}$ | 0.0 | 1090 |
| 9 [b] | $H+H\Leftrightarrow H_2+M$ | $6.40 \times 10^{17}$ | −1.0 | 0 |
| 10 [b] | $H+OH\Leftrightarrow H_2O+M$ | $2.20 \times 10^{22}$ | −2.0 | 0 |

[a] Units are in seconds, moles, cubic centimeters, calories, and degrees Kelvin. [b] Third-body efficiencies for all thermolecular reactions are 2.5 for M = $H_2$, 16.0 for $H_2O$, and 1.0 for all other M, respectively.

### 2.4. Boundary Conditions and Meshing

In the LOX/kerosene fuel-rich gas generator, the inlet temperatures of the fuel and oxidants were 300K for RP-3 kerosene and 91K for liquid oxygen, respectively. The pressure boundary condition was adopted at the outlet of the gas generator, and the outlet pressure was 101,325 Pa. The solid wall of the gas generator was treated as an adiabatic wall, and the near-wall flow is modeled by the standard wall function.

The computational model is divided into several blocks, and each block was meshed separately. The grid was drawn by Gambit2.4.6 software [30], as shown in Figure 3. The whole computational area was approximately 2.5 million grids, with an average grid size of around 1 mm.

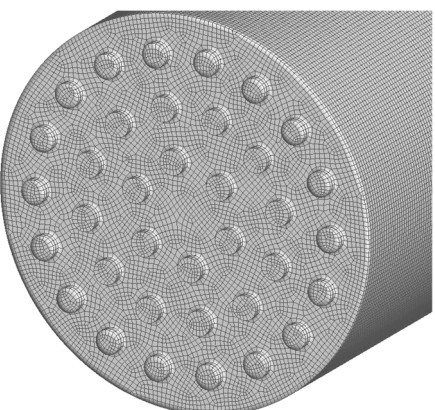

**Figure 3.** Grids of the computational domain.

### 2.5. Experiment Verification of Computation Method

To validate the numerical simulation method employed in this study, a fuel-rich combustion test of the $GO_2$/kerosene torch igniter was conducted. The combustion efficiency of the torch igniter was obtained through the $GO_2$/kerosene torch igniter experiment and compared with the numerical simulation results to verify the accuracy of the computational

methods employed in this paper. The numerical simulation in this paper was performed using ANSYS Fluent 2020R1 software [31], and the resulting contour data was processed using Tecplot360 EX2017 R3 software [32].

The torch igniter used was 194 mm in length, 32 mm in body diameter, 5.91 mm in outlet diameter, and 2.94 m in characteristic length, respectively and the characteristic length of the $GO_2$/kerosene torch igniter was close to that of the LOX/kerosene gas generator. The physical structure of the torch igniter is shown in Figure 4a.

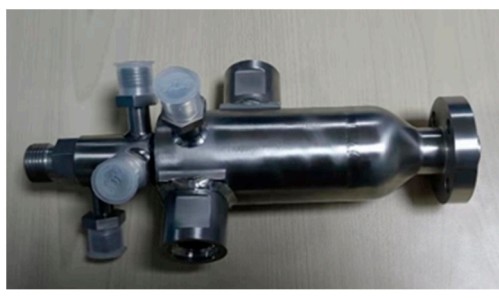

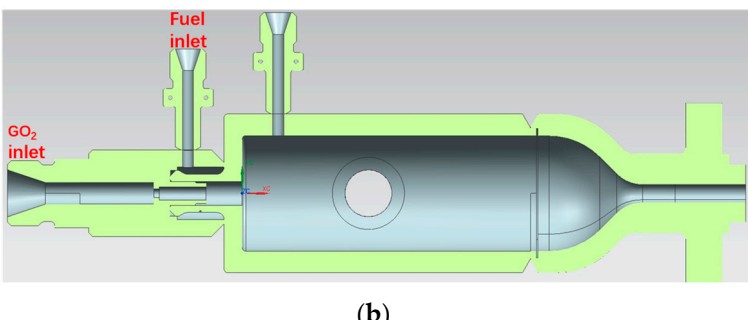

(**a**)                          (**b**)

**Figure 4.** Structural and section view of the igniter. (**a**) Structural of the torch igniter; and (**b**) cross-view of the torch igniter.

The $GO_2$ inlet is a plain orifice injector along the radial direction, while the kerosene inlet is a swirl injector along the axial direction. The plain orifice injector and the swirl injector form an internally mixed bi-component coaxial swirl injector, which is similar to the injector in the LOX/kerosene gas generator.

As the working time of the torch igniter is short, and the heat sink cooling is mainly used in the burners with a short working time, the torch igniter used in this paper adopted the heat sink cooling method with spark plug ignition. The cross-view of the igniter is shown in Figure 4b. The experiment mixing ratio and mass flow rate are shown in Table 4.

**Table 4.** Mixing ratios and mass flow rates of the torch igniter experiment.

| No. | Km | m (Kg/s) |
| --- | --- | --- |
| 1 | 0.454 | 0.1089 |
| 2 | 0.557 | 0.1118 |
| 3 | 0.599 | 0.0798 |
| 4 | 0.656 | 0.0959 |

In the $GO_2$/kerosene torch igniter experiment, $GO_2$ and kerosene were supplied by a high-pressure gas holder and piston pump, respectively. The ignition time series is shown in Figure 5, and the working time of the torch igniter was 4 s. The gas generator pressure was obtained through setting a pressure sensor on the body of the torch igniter. The image of the torch igniter wake at different times is shown in Figure 6.

Figure 6 is the slipstream of the $GO_2$/kerosene torch igniter at different times during the mixing ratio 0.599 test. It can be seen from Figure 6 how a flame was generated from the outlet of the torch igniter, which was due to the secondary combustion of the high-temperature fuel-rich gas and outside air. Figure 6 also shows that the slipstream flame resulting from the oxidation of soot in the fuel-rich gas appears as a bright yellow color. At t = 2.5 s, a local flameout phenomenon can be observed, indicated by the red dotted line in the graph. This occurs as the high-temperature fuel-rich gas was ejected at the local speed of sound, thereby creating a region where the local slipstream velocity exceeds the flame propagation velocity, leading to local flameout.

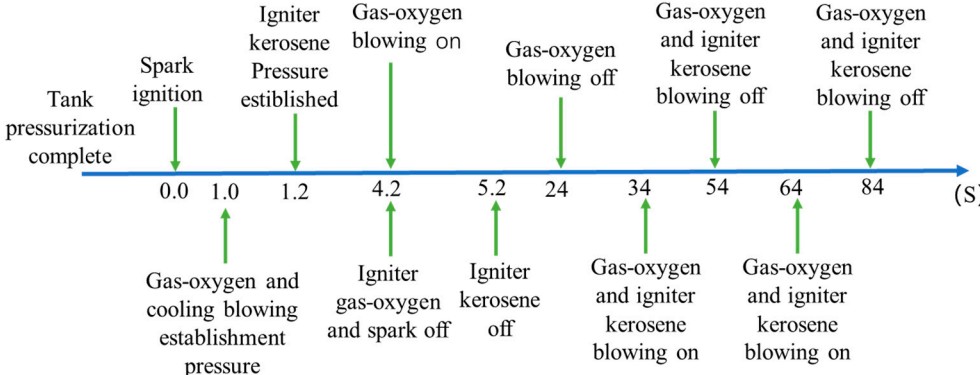

**Figure 5.** Time sequence of the ignition experiment.

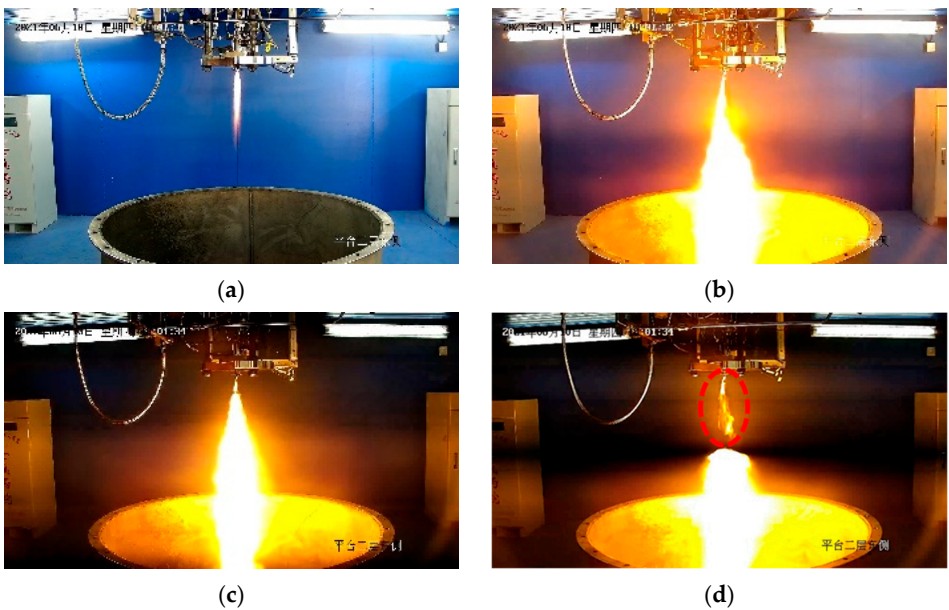

**Figure 6.** Slipstream of the torch igniter under Km = 0.599. (**a**) At t = 1.0 s; (**b**) t = 1.5 s; (**c**) t = 2.0 s; and (**d**) t = 2.5 s, respectively.

In the actual combustion chamber, where the nozzle throat area $A_t$ and the propellant mixing ratio is determined, due to the limitation of the design level, there are always a series of losses in the process of propellant conversion from chemical energy to thermal energy. In other words, the mass flow rate of the same propellant cannot obtain the total pressure of the combustion chamber. In order to evaluate the loss degree in the energy conversion process of the combustion chamber, the concept of combustion efficiency was introduced. Combustion efficiency refers to the ratio of the actual characteristic velocity $C_{ex}^*$ to the theoretical characteristic velocity $C_{th}^*$, specifically:

$$\eta_c = \frac{C_{ex}^*}{C_{th}^*} = \frac{P_c A_t}{\dot{m} C_{th}^*} \tag{4}$$

Figure 7 shows the comparison between the experimental results and the numerical simulation results of the torch igniter combustion efficiency under four conditions. The abscissa and ordinate are the numerical simulation results and experimental results of combustion efficiency under the four conditions, respectively. It can be seen from Figure 7, that in the three conditions of Km = 0.454, Km = 0.557, and Km = 0.599, respectively, the numerical simulation results of the combustion efficiency are slightly higher than the experimental results. In the condition of Km = 0.656, the numerical simulation results of the combustion efficiency are slightly lower than the experimental results, with the maximum relative error

(Relative error = (numerical efficiency − experimental efficiency)/experimental efficiency × 100%) of 1.41%, and the average error of 0.61%, and the error was within the acceptable range. Therefore, the computational method in this paper was deemed to be feasible.

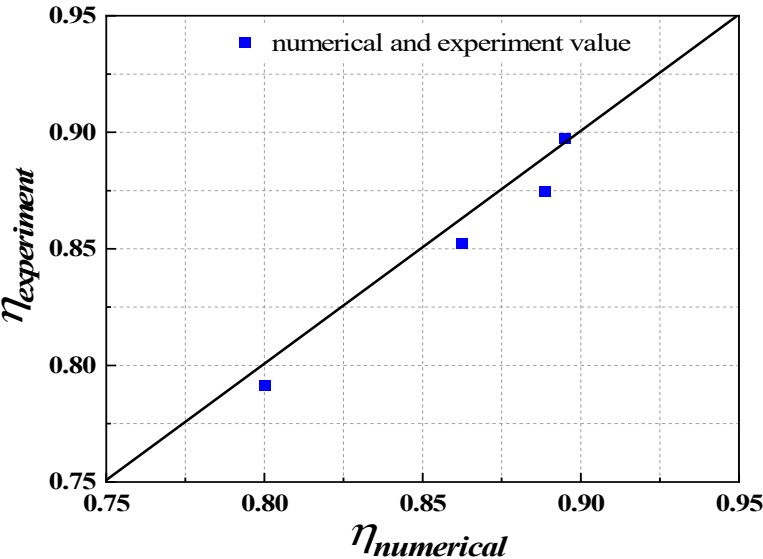

**Figure 7.** Comparison of the combustion efficiency obtained by numerical simulation and experimental results.

## 3. Results and Discussion

### 3.1. Flow Field and Performance Analysis of Different Local Mixing Ratios

#### 3.1.1. Effect of Different Local Mixing Ratios on Kerosene Concentration

Figure 8 shows the contour of kerosene concentration distribution in the symmetrical section of the gas generator under the condition of $m_{-1}$ = 0.10 kg/s ($m_{-2}$ = 0.10 kg/s) and different $Km_{-1}$. It can be seen from Figure 8 that the kerosene concentration near the wall of the gas generator was higher under different conditions, and the concentration gradually decreased along the flow direction. The main reason for this was attributed to the outer-ring nozzle of the gas generator being a monopropellant kerosene nozzle, and the ejected kerosene forming a low-temperature kerosene vapor film near the wall of the combustion chamber to reduce the wall temperature. Due to the kerosene from the outer-ring injector occurring turbulent diffusion with the propellant from the inner-ring and the middle-ring injector, partially cooled kerosene near the wall was involved in the combustion reaction, which caused the kerosene concentration near the wall area to gradually decrease along the flow direction. It can also be seen from Figure 8a–c that there is an area with a high kerosene concentration in the middle which reaches the central axis of No.1 to No.3 conditions. With the increase of the $Km_{-1}$ and decrease of the $Km_{-2}$, the kerosene concentration in the above area gradually decreased as a result. This is because the mixing ratio was lower, while the kerosene concentration was higher. When the mass flow rates of the inner-ring injector and the middle-ring injector were fixed, with the increase of the $Km_{-1}$ (the kerosene mass flow rate of the inner-ring injector decreases), and when below the stoichiometric ratio (when fuel and air are completely burned, the ratio of fuel quality to air quality), a slight increase in the mixing ratio was found to be more conducive to the combustion reaction, and in this area the combustion was more sufficient, leading to the gradual decrease in the kerosene concentration in the middle reaching the central axis. Furthermore, Figure 8d–e demonstrates that as Km-1 increases, there is a general decrease in the kerosene concentration in the middle reaching of the central axis, indicating increased kerosene participation in the combustion reaction.

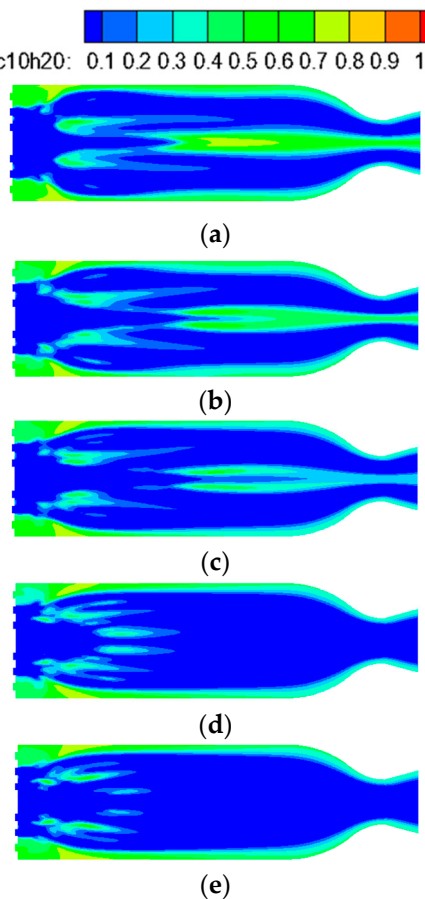

**Figure 8.** Kerosene distributions of y = 0 plane under different local mixing ratios. (**a**) No.1 ($Km_{-1}$ = 0.43, $Km_{-2}$ = 3.80); (**b**) No.2 ($Km_{-1}$ = 0.70, $Km_{-2}$ = 2.79); (**c**) No.3 ($Km_{-1}$ = 1.00, $Km_{-2}$ = 2.24); (**d**) No.4 ($Km_{-1}$ = 1.69, $Km_{-2}$ = 1.69); and (**e**) No.5 ($Km_{-1}$ = 2.33, $Km_{-2}$ = 1.45).

3.1.2. Effect of Different Local Mixing Ratios on the Oxygen Concentration

Figure 9 shows the contour of the oxygen concentration distribution in the symmetrical section of the gas generator under the condition of $m_{-1}$ = 0.10 kg/s ($m_{-2}$ = 0.10 kg/s) and different $Km_{-1}$. It can be seen from Figure 9 that along the axial direction, oxygen shows a diffused distribution after ejection from the inner-ring injector and the middle-ring injector. Impacted by the outer-ring cooling kerosene injector, the near-wall oxygen concentration distribution formed a complementary shape with the near-wall kerosene concentration distribution in Figure 8. Along the axis direction, the oxygen concentration near the injection panel of the gas generator was high, with oxygen participating in the combustion reaction, and the oxygen concentration downstream of the gas generator was found to gradually decrease. Along the radial direction, the area with high oxygen concentrations was mainly distributed in the downstream area of the inner-ring injector and the middle-ring injector. With the increase of $Km_{-1}$ (or the decrease of $Km_{-2}$), the oxygen concentration in the downstream area of the inner-ring injector and middle-ring injector gradually increased and decreased, respectively.

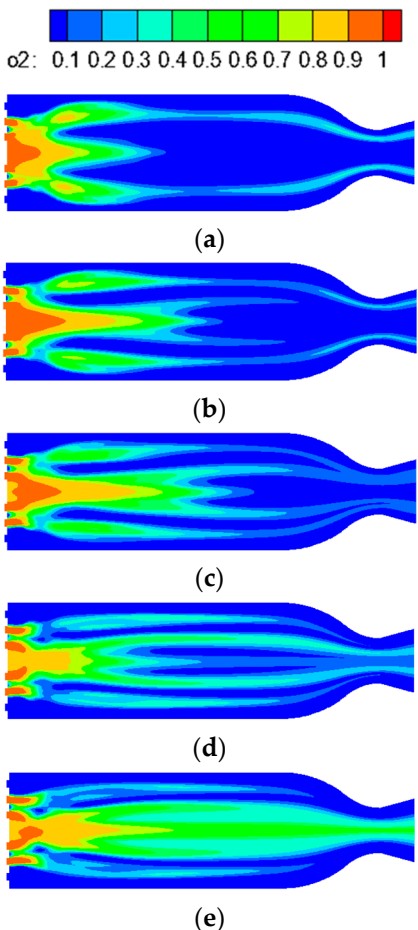

**Figure 9.** Oxygen distributions of y = 0 plane under different local mixing ratios. (**a**) No.1 ($Km_{-1}$ = 0.43, $Km_{-2}$ = 3.80); (**b**) No.2 ($Km_{-1}$ = 0.70, $Km_{-2}$ = 2.79); (**c**) No.3 ($Km_{-1}$ = 1.00, $Km_{-2}$ = 2.24); (**d**) No.4 ($Km_{-1}$ = 1.69, $Km_{-2}$ = 1.69); and (**e**) No.5 ($Km_{-1}$ = 2.33, $Km_{-2}$ = 1.45).

At the outlet of the gas generator, the mass flow average of oxygen concentration under No.1-No.5 conditions was 7.62%, 5.64%, 6.00%, 8.02%, and 9.58%, respectively. From the No.1 condition to the No.2 condition, with the increase of $Km_{-1}$ (or the decrease of $Km_{-2}$), the oxygen concentration at the outlet of the gas generator decreased, and reached the minimum under the No.2 condition, which in turn increased monotonously from the No.2 condition to the No.5 condition. Under the No.1 condition, the residual oxygen at the outlet of the gas generator was mainly from the middle-ring injector. During this time, the $Km_{-2}$ was relatively high ($Km_{-2}$ = 3.8), the oxygen content was excessive, and the combustion was found to be incomplete, resulting in the unburned oxygen being discharged from the outlet of the gas generator. When $Km_{-2}$ was decreased, the oxygen concentration downstream from the middle-ring was reduced in the No.2 condition while $Km_{-1}$ was increased, and the oxygen concentration of the inner-ring injector downstream was found to have increased significantly. Nevertheless, the oxygen concentration was still low under this condition, the oxygen of the inner-ring injector was able to be completely reacted, and the unburned oxygen at the gas generator outlet was mainly emitted by the middle-ring injector, but the outlet concentration was found to be significantly lower than the No.1 condition. From the No.2 condition to the No.5 condition, as $Km_{-1}$ increased and $Km_{-2}$ decreased, it was clear that the oxygen concentration downstream from the middle-ring injector gradually decreased, while the oxygen concentration of the inner-ring injector downstream gradually increased. At the outlet of the gas generator, the residual oxygen was mainly determined by the inner-ring injector.

The lower oxygen concentration at the gas generator outlet represents the higher combustion efficiency. This is because when the combustion state is fuel-rich combustion, the outlet residual oxygen concentration lowers, more oxygen is involved in the combustion process, and the combustion reaction becomes more complete, resulting in the observed higher combustion efficiency.

### 3.1.3. Effect of Different Local Mixing Ratios on Temperature Change

Figure 10 shows the average temperature profiles of the cross-section in the different axial distances from the injection plane under the condition of $m_{-1}$= 0.10 kg/s ($m_{-2}$= 0.10 kg/s) and different $Km_{-1}$. From Figure 10, it can be seen that the temperature rose relatively fast in the range of z = 0–120 mm, and from z = 120 mm to the outlet the temperature change was found to be relatively small. This is because in the range of z = 0–120 mm, kerosene and oxygen were rapidly injected and diffused into the gas generator through the injector, and the flow velocity of kerosene and oxygen was fast, while the concentration of kerosene and oxygen near the injection panel domain was high, indicating that as the turbulence disturbance was strong and the chemical reaction rate was fast, the temperature rise was fast as a result. As the reaction proceeded, the mass flow of oxygen and kerosene decreased, and then the combustion reaction rate decreased, and the combustion heat release rate decreased, which caused the temperature rise rate to change slowly. The concentration of oxygen and kerosene near the outlet was found to be lower, the temperature rise rate was further reduced, and the temperature curve was deemed to be basically unchanged.

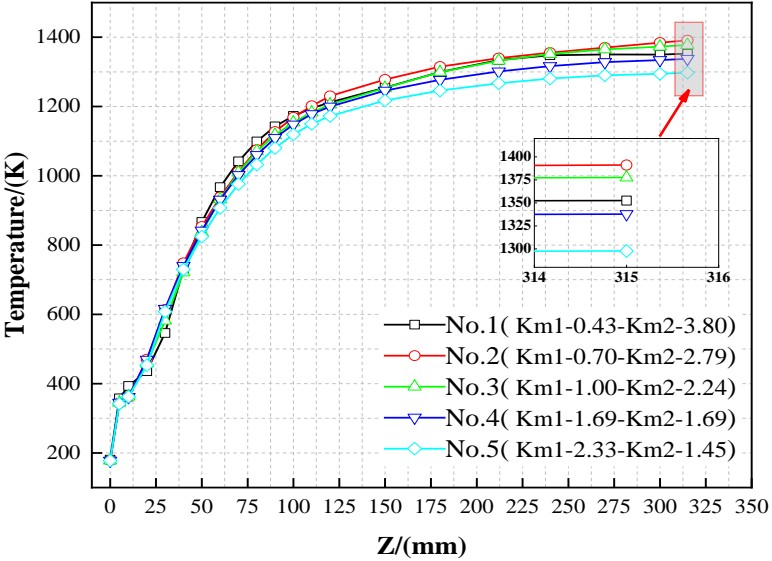

**Figure 10.** Temperature profiles under different local mixing ratios.

It can be seen from Figure 10 that the highest average outlet temperature was under the No.2 condition ($Km_{-1}$ = 0.70; $Km_{-2}$ = 2.79), reaching 1391 K, while the lowest average outlet temperature was 1297 K under the No.5 condition ($Km_{-1}$ = 2.33; $Km_{-2}$ = 1.45). Compared with Figure 9, the variation in the outlet temperature of the gas generator was opposite to the variation of the outlet oxygen concentration. This is because the lower outlet residual oxygen concentration represented the more oxygen involved in the combustion reaction, and the complete combustion reaction was higher. Therefore, the lower the outlet oxygen concentration, the higher the outlet temperature, and vice versa.

Figure 11 shows the temperature distribution and velocity streamline diagram in the symmetry plane of the gas generator under the condition of $m_{-1}$ = 0.10 kg/s ($m_{-2}$ = 0.10 kg/s) and different $Km_{-1}$. It can be seen from Figure 11 that the velocity streamlines of the gas generator are almost the same under the different conditions, there are 'pair vortex' and

'corner vortex' in the center position of the injection plane downstream and the edge domain of the gas generator. Compared with Figure 9, it can be seen that the oxygen concentration was higher in the central "pair vortex" area, while the kerosene concentration was higher in the boundary "corner vortex" area, thereby indicating a low temperature. According to the temperature distribution of the other areas, it can be found that the temperature near the injector panel was low, while the temperature gradually increased in the middle and downstream domains of the gas generator until the temperature at the outlet reached its highest. This occurs due to the incomplete mixing of 300 K kerosene and 90 K oxygen in the downstream area of the injector, resulting in insufficient chemical reactions and lower temperatures. As the propellant flows downstream, the mixing of kerosene and oxygen became more homogeneous, leading to a more efficient reaction and a gradual increase in combustion temperature. Eventually, the highest combustion temperature was reached at the outlet of the gas generator. Meanwhile, another low-temperature area was found to exist along the central axis and near the wall of the gas generator, while the temperature between the central axis and the adjacent wall (within approximately 50% of the radius range) was found to be higher. This phenomenon can be attributed to the relatively higher oxygen concentration in the central axis region and the higher kerosene concentration near the wall area, resulting in the formation of two lower-temperature regions within the gas generator. From the point of view of gas generator design, the lower temperature area of the gas generator distributed near the injector panel and near the body wall can prevent ablation damage to the injection panel and body wall.

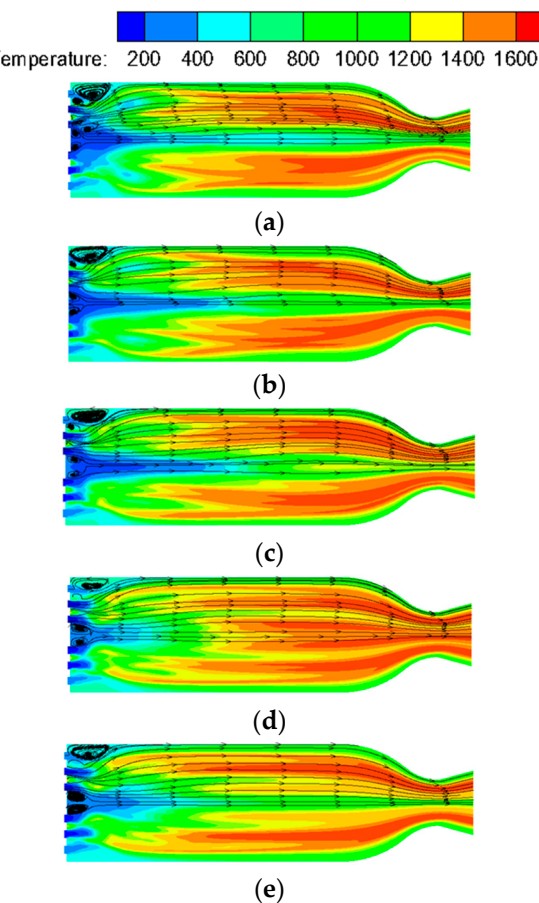

**Figure 11.** Temperature distributions of y = 0 plane under different local mixing ratios. (**a**) No.1 ($Km_{-1}$ = 0.43, $Km_{-2}$ = 3.80); (**b**) No.2 ($Km_{-1}$ = 0.70, $Km_{-2}$ = 2.79); (**c**) No.3 ($Km_{-1}$ = 1.00, $Km_{-2}$ = 2.24); (**d**) No.4 ($Km_{-1}$ = 1.69, $Km_{-2}$ = 1.69); and (**e**) No.5 ($Km_{-1}$ = 2.33, $Km_{-2}$ = 1.45).

It can also be seen in Figure 11a–c that with the increase of $Km_{-1}$ from 0.43 to 1 (or the decrease of $Km_{-2}$ from 3.8 to 2.24), there was a slight increase in temperature observed in the high-temperature region at a 50% radius, which was deemed to be influenced by the change in $Km_{-2}$. Meanwhile, the low-temperature area near the head of the gas generator center was also found to have increased, which was influenced by the change in $Km_{-1}$. Compared with Figure 9, it can be seen that with the decrease of $Km_{-2}$, the oxygen concentration in the downstream flow field of the middle-ring injector gradually decreased. As the kerosene is over-rich under all examined conditions, the decrease in oxygen concentration indicates that the combustion was more sufficient, resulting in a higher local temperature. Similarly, the increase in $Km_{-1}$ led to a gradual increase in the oxygen concentration in the central axial region of the gas generator, indicating an insufficient oxygen reaction in this region. Additionally, due to the extremely low injection temperature of oxygen (90 K), the low-temperature area near the head region of the gas generator was found to gradually expand.

It can be found in Figure 11d that when the $Km_{-1} = Km_{-2} = 1.69$, the temperature in the central axial area significantly increased, meaning the temperature distribution in the gas generator was relatively uniform. This is because the mixing ratios of all the combustion injectors were 1.69, while the distributions of oxidants and fuel in the whole gas generator were relatively uniform, resulting in a relatively uniform temperature distribution inside the gas generator.

When the $Km_{-1}$ was from 1.69 to 2.33 (or the $Km_{-2}$ from 1.69 to 1.45), it can be seen in Figure 11e that there is a low-temperature area (of approximately 1000 K) in the central area of the gas generator. The specific reason can be seen from Figure 9e: when the $Km_{-1}$ increased to 2.33, there was a large amount of incomplete combustion oxygen observed in the central area of the gas generator, resulting in a low local temperature.

Figure 12 reveals the contour of the temperature distribution in the cross-section at z = 210 mm of the gas generator under the condition of $m_{-1} = 0.10$ kg/s ($m_{-2} = 0.10$ kg/s) and different $Km_{-1}$. The black circles represent the relative positions of the injector. It can be seen from Figure 12 that the temperature of the cross-section at z = 210 mm was of a layered distribution, and that the local temperature was closely related to the mixing ratio of the local injector. Due to the lower $Km_{-1}$ in the No.1 condition ($Km_{-1} = 0.43$; $Km_{-2} = 3.80$), the combustion reaction was deemed to be insufficient, leading to a lower temperature at the center of the cross-section at z = 210 mm (below 1000 K). There was a ring-shaped high-temperature area observed at the location of the middle-ring injector (approximately 50% of the radius range), which was formed as a result from the combined effect of the middle-ring injector and the inner-ring injector. This is due to the fact that in the No.1 condition the $Km_{-2}$ was 3.80 (oxygen-enriched), which is close to the stoichiometric ratio of the oxygen and kerosene reaction, meaning the combustion reaction is more sufficient, resulting in the temperature of the middle-ring injector being higher. Meanwhile, in the No.1 condition, the $Km_{-1}$ was 0.43 and the kerosene concentration was too high, meaning that the excess oxygen in the middle-ring reacts with the excess kerosene in the inner-ring, causing the generation of a high-temperature area in the inner-ring injector downstream. Since the outer ring injector is a monopropellant kerosene injector, the species of the area near the wall was mainly kerosene, and in these areas the temperature was about 800 K and was relatively low at different mixing ratios, meaning therefore the low-temperature kerosene lays a good foundation for the reliable work of the gas generator. As part of the oxygen in the middle-ring injector will react with the kerosene near the wall of the gas generator through turbulent diffusion, this causes the generation of the high-temperature area in the downstream flow field between the outer-ring injector and the middle-ring injector. The dimension of the high-temperature domain and the maximum temperature were found to be significantly lower than those of the gas generator at the 50% radius.

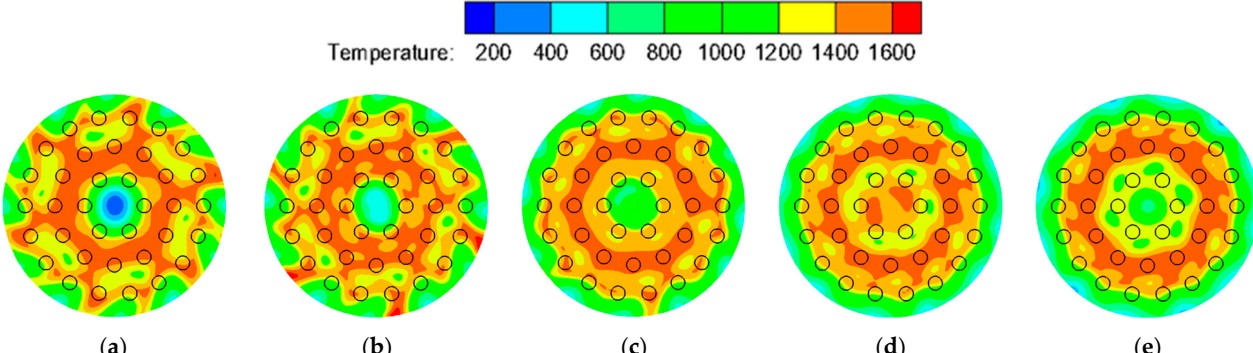

**Figure 12.** Temperature distributions of z = 210 mm cross-section under different local mixing ratios. (**a**) No.1 ($Km_{-1}$ = 0.43, $Km_{-2}$ = 3.80); (**b**) No.2 ($Km_{-1}$ = 0.70, $Km_{-2}$ = 2.79); (**c**) No.3 ($Km_{-1}$ = 1.00, $Km_{-2}$ = 2.24); (**d**) No.4 ($Km_{-1}$ = 1.69, $Km_{-2}$ = 1.69); and (**e**) No.5 ($Km_{-1}$ = 2.33, $Km_{-2}$ = 1.45).

From Figure 12b–d, it can be seen that with increases of $Km_{-1}$ (or the decrease of $Km_{-2}$), the temperature in the inner-ring injector downstream of the gas generator significantly increased, while the high temperature area located downstream of the middle-ring injector decreased gradually, indicating that the cross-sectional temperature of the gas generator was uniform. The main reason attributed to this was that with the increase of $Km_{-1}$ and the decrease of $Km_{-2}$, the combustion in the central area was more sufficient, while the fuel in the 50% radius area was further enriched, leading to the increase and decrease of the temperature in the central area and the 50% radius area, respectively. As a result, the cross-sectional temperature of the gas generator (except for the edge area) tended to be uniform.

Figure 12e reveals that when $Km_{-1}$ reaches 2.33, a significant amount of residual oxygen was found to accumulate in the center of the gas generator, leading to an extremely insufficient combustion reaction and a decrease in temperature in this region. However, a discrepancy arises when comparing Figure 12e with Figure 12c. In the No.5 condition, where $Km_{-1}$ = 2.33, and the No.3 condition, where $Km_{-2}$ = 2.24, the values of these two mixing ratios are relatively similar. However, the temperature in the center of the No.5 condition was found to be significantly lower than that in the 50% radius area of the No.3 condition. This disparity may be attributed to the fact that the flow and the local mixing ratio in the central area of the gas generator were primarily influenced by the inner-ring injector, whereas the flow state and the local mixing ratio in the 50% radius area of the gas generator were jointly influenced by the strong shear forces between the inner, middle, and outer ring injectors, leading to a more intense turbulent combustion. Therefore, the combustion of the middle-ring injector was deemed to be more sufficient, and the temperature was recorded to be higher. The reason that the oxygen residual first decreases and then increases with the change of mixing ratio has been previously discussed in the analysis of Figure 9, which is not repeated here.

### 3.1.4. Effect of Different Local Mixing Ratios on Combustion Efficiency

Figure 13 reveals the variation of combustion efficiency under the condition of $m_{-1}$ = 0.10 kg/s ($m_{-2}$= 0.10 kg/s) and different $Km_{-1}$. The numbers near the data points in Figure 13 represent the condition numbers, with the same number indicating the same condition. From Figure 13, it is evident that as $Km_{-1}$ increases (or $Km_{-2}$ decreases), the combustion efficiency of the gas generator increases from condition No.1 to No.2 and decreases from condition No.2 to No.5. The combustion efficiency reaches its peak value of $\eta$ = 89.10% when $Km_{-1}$ = 0.70 and $Km_{-2}$ = 2.79 in condition No.2, respectively. The combustion efficiency can be represented with the residual oxygen concentration of the gas generator outlet by comparing Figure 9 with Figure 13, where from No.1 to No.5 conditions in Figure 9, the oxygen concentration at the outlet of the gas generator initially decreases and then increases, and reaches the lowest value under the No.2 condition (with outlet oxygen

concentration of 5.64%), which is consistent with the results shown in Figure 13. A lower residual oxygen content indicates more complete combustion and a higher combustion efficiency. Conversely, a higher oxygen excess leads to a lower combustion efficiency.

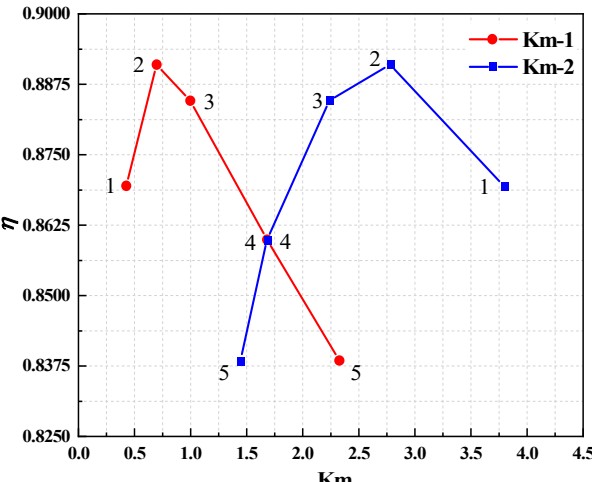

**Figure 13.** Combustion efficiency under different local mixing ratios.

### 3.2. Flow Field and Performance Analysis of Different Local Flow

#### 3.2.1. Effect of Different Local Mass Flow Rates on Kerosene Concentration

Figure 14 shows the contour of the kerosene concentration distribution in the symmetrical section of the gas generator under the condition of $Km_{-1} = Km_{-2} = 1.69$ and different $m_{-1}$. According to Figure 14, the kerosene concentration was higher near the wall of the gas generator, and with the increase of $m_{-1}$ and the decrease of $m_{-2}$, the kerosene concentration near the wall remained unchanged. This result is relevant in relation to the main purpose of our study to change the mass flow rates and mixing ratios to study the combustion flow field and the efficiency of the gas generator, without considering the effects of changing local parameters on the wall temperature. In order to ensure the feasibility of the conclusions drawn in this paper, when changing the mass flow rates and mixing ratios of the inner- and middle-ring injectors, the mass flow rate of the out-ring injector was kept as the unchanged value to ensure the reliability of the gas generator wall cooling. Consequently, the kerosene concentration near the wall area in Figure 14a–e remained relatively constant. In addition, it can be seen from Figure 14 that with the increase of $m_{-1}$ and the decrease of $m_{-2}$, the kerosene concentration in the downstream area corresponding to the inner-ring injector of the gas generator gradually increased, while the kerosene concentration in the downstream area corresponding to the middle-ring injector gradually decreased. This is because the mixing ratio of the inner-ring injector and the middle-ring injector was consistent, and the combustion reaction was fuel-rich, and in this condition, injectors with larger mass flow rates correspond to larger kerosene mass flow rates. These local areas retained a higher concentration of kerosene in the fuel-rich combustion, meaning the kerosene concentration in these areas was higher.

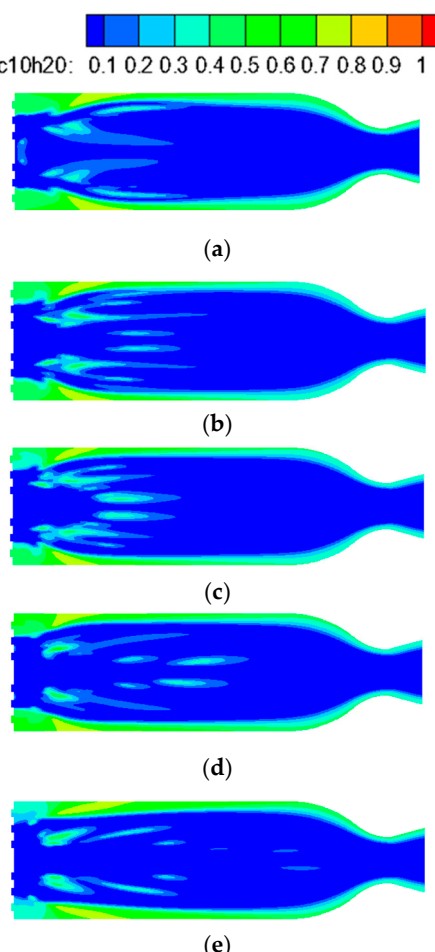

**Figure 14.** Kerosene distributions of y = 0 plane under different local mass flow rates. (**a**) No.6 ($m_{-1}$ = 0.02 kg/s, $m_{-2}$ = 0.14 kg/s); (**b**) No.7 ($m_{-1}$ = 0.06 kg/s, $m_{-2}$ = 0.12 kg/s); (**c**) No.8 ($m_{-1}$ = 0.10 kg/s, $m_{-2}$ = 0.10 kg/s); (**d**) No.9 ($m_{-1}$ = 0.14 kg/s, $m_{-2}$ = 0.08 kg/s); and (**e**) No.10 ($m_{-1}$ = 0.20 kg/s, $m_{-2}$ = 0.05 kg/s).

### 3.2.2. Effect of Different Local Mass Flow Rates on Oxygen Concentration

Figure 15 shows the contour of the oxygen concentration distribution in the symmetrical section of the gas generator under the condition of $Km_{-1} = Km_{-2} = 1.69$ and different $m_{-1}$. It can be seen from Figure 15 that the oxygen concentration distribution at the injector downstream has a relatively obvious consistency with the mass flow rates of the inner and middle injectors. With the increase of $m_{-1}$ and the decrease of $m_{-2}$, the oxygen concentration downstream of the inner-ring injector and the middle-ring injector were increased and decreased, respectively, the main reason being the mixing time of kerosene and oxygen when the structure size of the injector was fixed, increasing the propellant mass flow rate that will inevitably increase the propellant injection velocity, and thereby reducing the residence time of the propellant in the gas generator, As a result, the local combustion reaction becomes insufficient, leading to an increase in the local oxygen concentration.

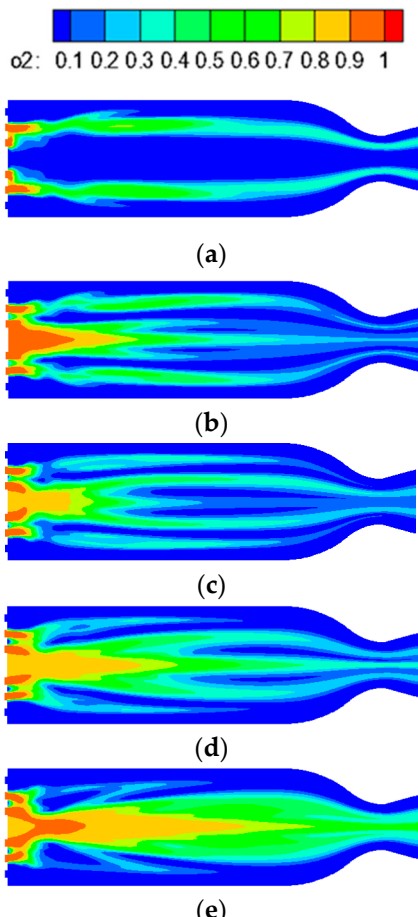

**Figure 15.** Oxygen distribution of y = 0 under different local mass flow rates. (**a**) No.6 ($m_{-1}$ = 0.02 kg/s, $m_{-2}$ = 0.14 kg/s); (**b**) No.7 ($m_{-1}$ = 0.06 kg/s, $m_{-2}$ = 0.12 kg/s); (**c**) No.8 ($m_{-1}$ = 0.10 kg/s, $m_{-2}$ = 0.10 kg/s); (**d**) No.9 ($m_{-1}$ = 0.14 kg/s, $m_{-2}$ = 0.08 kg/s); and (**e**) No.10 ($m_{-1}$ = 0.20 kg/s, $m_{-2}$ = 0.05 kg/s).

At the outlet of the gas generator, the mass flow average of oxygen concentration under No.6 to No.10 conditions was 10.48%, 8.66%, 8.02%, 8.34%, and 12.46%, respectively. When the propellant mass flow rate of the inner and middle ring injectors was equal ($m_{-1}$ = $m_{-2}$ = 0.1 kg/s), the outlet oxygen concentration of gas generator reached its lowest value while the combustion was the most sufficient. When the total mass flow rate of gas generator remained unchanged, whether the $m_{-1}$ increased ($m_{-2}$ decreased) or $m_{-2}$ increased ($m_{-1}$ decreased), the residual oxygen at the outlet of the gas generator increased as a result. It can be seen from this result that the changed $m_{-1}$ or $m_{-2}$ in the gas generator was not consistent with the outlet oxygen concentration. The outlet oxygen residual directly reflects the combustion efficiency of the gas generator, which was closely related to the overall combustion flow field of the gas generator. The specific reason was be further described in combustion efficiency under different local mass flow rates -.

### 3.2.3. Effects of Different Local Mass Flow Rates on Exhaust Temperature

Figure 16 shows the average temperature profiles of the cross-section in the different axial distances from the injection plane under the condition of $Km_{-1}$ = $Km_{-2}$ = 1.69 and different $m_{-1}$. According to Figure 16, the temperature rose rapidly at z = 0–120 mm, and the slope of the curve was large. From z = 120 mm to the outlet, the temperature rise rate then decreased gradually, and the temperature rise curve was gentler. The reason for this was similar to the observations made of Figure 10, whereby as both the kerosene and oxygen concentrations near the injection panel were high, this is conducive to the

rapid combustion reaction. With the combustion reaction proceeding, the concentrations of kerosene and oxygen gradually decreased, causing the combustion reaction rate and the temperature rise rate to decrease.

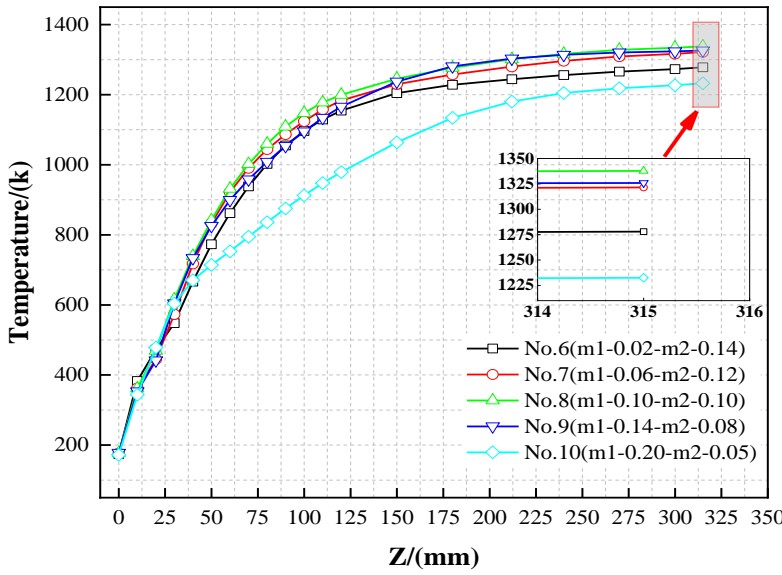

**Figure 16.** Temperature profiles under different local mass flow rates.

The average outlet temperatures of the gas generator under the No.6 to No.10 conditions were 1277.97 K, 1321.48 K, 1337.76 K, 1325.86 K, and 1232.49 K, respectively. This is consistent with the variation of oxygen residues observed in Figure 15. As the gas generator operates under a fuel-rich LOX/kerosene combustion reaction, with the same total mass flow rate and total mixing ratio, a lower oxygen concentration at the gas generator outlet indicates more complete combustion. Consequently, the average outlet temperature of the gas generator was higher in this context.

Figure 17 shows the temperature distribution and velocity streamline diagram in symmetry plane of gas generator under the condition of $Km_{-1} = Km_{-2} = 1.69$ and different $m_{-1}$. It can be seen from Figure 17 that in the symmetry plane of the gas generator the high-temperature area mainly has two areas: the center axial area and near the wall area (about 80% radius position), and the two high-temperature areas are mainly affected by the inner-ring injector and the middle-ring injector, respectively. With the increase of $m_{-1}$ and the decrease of $m_{-2}$, it can be clearly seen that the high-temperature area near the center of the gas generator gradually decreases and moves downstream. It can be further seen in Figure 17a that the temperature of the center axial area of the gas generator was higher under the No.6 condition, especially near the injection panel, and the temperature was close to 1600 K. This was mainly due to the $m_{-1}$ in the No.6 condition, whereby the longer residence time of the propellant in the gas generator represented the more sufficient combustion reaction, resulting in a higher temperature in the center axial area. In addition, due to the $m_{-2}$ being large and the $m_{-1}$ being small in the No.6 condition, the difference in mass flow rates led to a recirculation area between the inner-ring injector and the middle-ring injector, and the existence of a recirculation area greatly increased the residence time of the propellant, meaning the combustion is more sufficient, and the combustion temperature is increased as a result. Generally, it is advisable to avoid high-temperature areas near the injection panel as they can cause injector ablation and significantly affect the reliable operation of the injector. From Figure 17b–e, the effect of changing the $m_{-1}$ and the $m_{-2}$ on the temperature distribution of gas generator was found to be complementary to the effect of Figure 15b–e on the oxygen concentration distribution. This occurs as a higher mass flow rate in the inner ring injector results in a higher local flow rate, thereby reducing the residence time of oxygen and kerosene in the gas generator. As a result, the combustion

reaction becomes less complete, leading to a higher amount of residual oxygen. Therefore, compared with Figures 15b–e and 17b–e, where the higher the oxygen concentration, the lower the local temperature distribution.

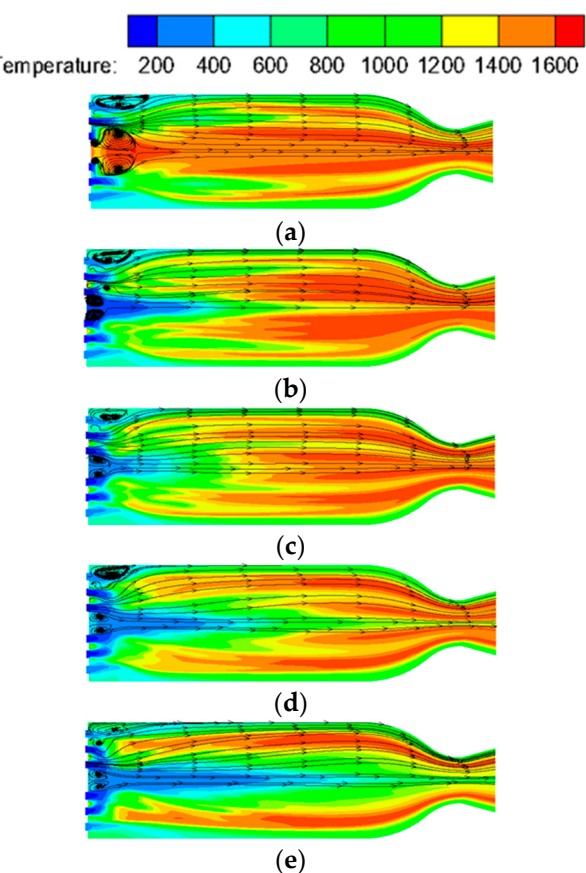

**Figure 17.** Temperature distributions of y = 0 plane under different local mass flow rates. (**a**) No.6 ($m_{-1}$ = 0.02 kg/s, $m_{-2}$ = 0.14 kg/s); (**b**) No.7 ($m_{-1}$ = 0.06 kg/s, $m_{-2}$ = 0.12 kg/s); (**c**) No.8 ($m_{-1}$ = 0.10 kg/s, $m_{-2}$ = 0.10 kg/s); (**d**) No.9 ($m_{-1}$ = 0.14 kg/s, $m_{-2}$ = 0.08 kg/s); and (**e**) No.10 ($m_{-1}$ = 0.20 kg/s, $m_{-2}$ = 0.05 kg/s).

Figure 18 shows the contour of the temperature distribution in the cross-section at z = 210 mm of the gas generator under the condition of $Km_{-1}$ = $Km_{-2}$ = 1.69 and different $m_{-1}$. The black circles in Figure 18 are the arrangement position of the inner, middle, and outer ring injectors at z = 0 mm. It can be seen from Figure 12 that the temperature of the cross-section at z = 210 mm was of a layered distribution, and that the local temperature was closely related to the mixing ratio of the local injector. Similar to Figure 12, the temperature at the cross-section at z = 210 mm was of a layered distribution, while the local temperature was found to be closely related to the mass flow rate of the local injector. In Figure 18a, there are two high-temperature areas in the cross-section: the central circular high-temperature area and the annular high-temperature area at about 50% radius position, which is affected by the mass flow rates of the inner-ring injector and middle-ring injector, respectively. This is due to the combustion reaction being more sufficient at the downstream domain corresponding to the injector. Due to the effects from the cooled kerosene from the outer-ring injector, the actual mixing ratio of the middle-ring injector downstream will be lower than 1.69, and therefore, the temperature of the annular high-temperature area was slightly lower than the central circular high-temperature area. From Figure 18a,b, it can be seen that with the increase of the $m_{-1}$ and the decrease of the $m_{-2}$, the area of the central circular high-temperature area was enlarged as a result, while the maximum temperature of the annular high-temperature area at 50% radius was found to have decreased. When the

$m_{-1} = m_{-2} = 0.10$ kg/s, the central circular high-temperature area and the annular high-temperature area at 50% radius merged, and a relatively uniform temperature distribution was formed at the cross-section of the gas generator as a result. This is primarily because the injectors in the inner and middle rings have the same mass flow rates, and the local mass flow rates in the gas generator are similar. As a result, the residence time of propellants in the gas generator is consistent, leading to a relatively uniform combustion reaction in the inner and middle rings. From Figure 18c–e, it can be seen that with the further increase of the $m_{-1}$ and the further decrease of the $m_{-2}$, the temperature of the central circular high-temperature area was significantly reduced, while the temperature of the annular high-temperature area at 50% radius was increased. The reason for these observations is that the increase of $m_{-1}$ (or decrease of $m_{-2}$) causes local flow velocity changes in the injection downstream domain, and as a result, the residence time of the propellant in the central circular high-temperature area and the annular high-temperature area at 50% radius decreased and increased, respectively. The longer the residence time of the propellant in the gas generator leads to the combustion being more sufficient, resulting in the higher temperature.

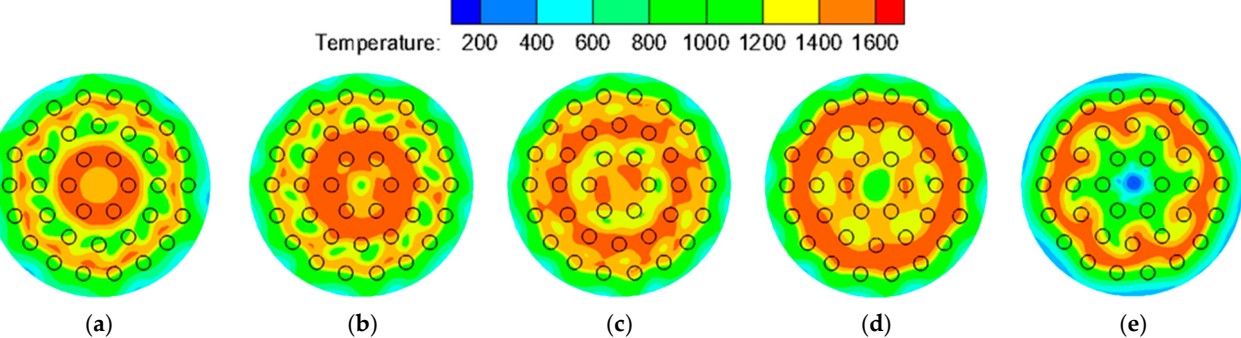

**Figure 18.** Temperature distributions of z = 210 mm cross-section under different local mass flow rates. (**a**) No.6 ($m_{-1} = 0.02$ kg/s, $m_{-2} = 0.14$ kg/s); (**b**) No.7 ($m_{-1} = 0.06$ kg/s, $m_{-2} = 0.12$ kg/s); (**c**) No.8 ($m_{-1} = 0.10$ kg/s, $m_{-2} = 0.10$ kg/s); (**d**) No.9 ($m_{-1} = 0.14$ kg/s, $m_{-2} = 0.08$ kg/s); and (**e**) No.10 ($m_{-1} = 0.20$ kg/s, $m_{-2} = 0.05$ kg/s).

3.2.4. Effects of Different Local Mass Flow Rates on Combustion Efficiency

Figure 19 shows the variation of combustion efficiency under the condition of $Km_{-1} = Km_{-2} = 1.69$ and different $m_{-1}$. The number next to each data point in the figure represents the condition number, and the same number represents the same conditions. It can be seen from Figure 19 that the combustion efficiency of a gas generator under different conditions was more than 80%. With the increase of $m_{-1}$ (or the decrease of $m_{-2}$), in the conditions from No.6 to No.8, the combustion efficiency of the gas generator was found to have increased, and in the conditions from No.8 to No.10, the combustion efficiency of the gas generator was found to have decreased, and the combustion efficiency reaches its maximum value of $\eta = 86.13\%$ under the No.8 condition ($m_{-1} = m_{-2} = 0.10$ kg/s). This indicates that a more uniform flow rate of the propellant in the inner-ring injector and the middle-ring injector promotes the combustion reaction. Under the condition where the total propellant flow is unchanged, the change of flow distribution will cause a change in the local mass flow rate, thus affecting the local mass flow rate and the residence time of the propellant in the gas generator, resulting in an insufficient combustion of the propellant and a reduced combustion efficiency.

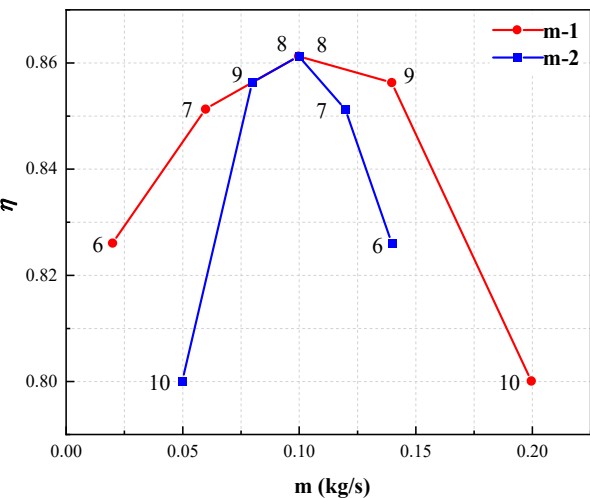

**Figure 19.** Combustion efficiency under different local mass flow rates.

It can also be seen from Figure 19 that in the No.6 condition where the $m_{-2}$ = 0.14 kg/s, and in the No.9 condition where the $m_{-1}$ = 0.14 kg/s, the combustion efficiency of the No.9 condition is higher than the No.6 condition. This is because the middle-ring injector has a larger location radius, and the mixing and diffusion are therefore more uniform in the whole gas generator. On the other hand, the inner-ring injector has a smaller radial position, resulting in less uniform mixing and diffusion compared to the middle-ring injector. Therefore, although the mass flow rates of the middle-ring injector in the No.6 condition are the same as the inner-ring injector in the No.9 condition, the combustion efficiency of the No.9 condition was higher.

## 4. Conclusions

This paper employs the numerical simulation method to analyze the combustion flow field and performance of a LOX/kerosene fuel-rich gas generator for the ATR engine. By varying the mass flow rate and mixing ratio of the individual injector, the flow and combustion characteristics of the gas generator were analyzed while maintaining a constant total mass flow rate and total mixing ratio. The main conclusions are as follows:

(1) As the mixing ratio in the inner-ring injectors increases (while the mixing ratio in the middle-ring injectors decreases), the oxygen concentration area near the axis zone and the 50% radius zone of the gas generator expands. Conversely, the kerosene concentration area near the axis zone decreases while gradually increasing near the 50% radius zone.

(2) In the flow direction section, there was an inverse relationship found between the variation trend of local temperature and the oxygen concentration in the local area. As the oxygen concentration increases, the temperature decreases.

(3) The temperature distribution across the cross-section of the gas generator exhibits a circular pattern. When the mixing ratio (or mass flow rates) of the unit injector are perfectly balanced, the temperature distribution becomes highly uniform. A larger disparity in the flow rate between the inner-ring injector and the middle-ring injector led to a lower combustion efficiency. This effect differs from the observed effect of the mixing ratio difference between the two injector rings.

(4) Increasing the mixing ratio in the inner-ring injectors (or decreasing the mixing ratio in the middle-ring injectors) initially led to a rise in the combustion efficiency, followed by a subsequent decline. The maximum combustion efficiency of 89.10% was achieved when the mixing ratio was set to $Km_{-1}$ = 0.7 and $Km_{-2}$ = 2.79, respectively. Increasing the flow rate in the inner-ring injectors (or decreasing the flow rate in the middle-ring injectors) also initially led to an improvement in the combustion efficiency, followed

by a subsequent decline. The maximum combustion efficiency of 86.13% was achieved when the mass flow rate was set to $m_{-1} = m_{-2} = 0.1$ kg/s.

In future developments, the analysis of different injector arrangements and varying numbers of injectors will be conducted. Additionally, the regenerative cooling will be calculated for the different propellant flow rates to ensure that the wall cooling meets the design requirements, and the combustion flow field and combustion characteristics of the gas generator according to the combustion instability (combustion oscillation) will be further investigated.

**Author Contributions:** All authors discussed and agreed upon the idea, and made scientific contributions: Conceptualization, Y.Z., B.H. and Q.Z.; Numerical simulation, Y.Z. and B.H.; Experiment, Y.Z., Q.S. and X.X.; Data analysis, Y.Z. and B.H.; Investigation, X.X. and W.Z.; Resources, W.Z. and Q.Z.; Supervision, Q.Z.; Writing—original draft, Y.Z. and B.H.; Writing—review and editing, Y.Z., B.H. and Q.S. All authors have read and agreed to the published version of the manuscript.

**Funding:** This research received no external funding.

**Data Availability Statement:** Due to privacy restrictions, we cannot share our research data.

**Conflicts of Interest:** The authors declare no conflict of interest.

## Nomenclature

| | |
|---|---|
| LOX | Liquid oxygen |
| $GO_2$ | Gas oxygen |
| ATR | Air turbo rocket |
| No. | Number |
| $Km_{-1}$ | Stoichiometric oxygen-to-fuel ratio in inner-ring injectors |
| $Km_{-2}$ | Stoichiometric oxygen-to-fuel ratio in middle-ring injectors |
| $m_{-1}$ | Mass flow of inner-ring injectors |
| $m_{-2}$ | Mass flow of middle-ring injectors |
| $\eta_c$ | Combustion efficiency |
| $k$ | Turbulent kinetic energy |
| $\varepsilon$ | Turbulent dissipation rate |
| $\rho$ | Density |
| $\mu$ | Dynamic viscosity |
| $\mu_t$ | Turbulent viscosity |
| $\gamma$ | Kinematic viscosity |
| $S_{ij}$ | Average strain rate |
| $G_k$ | Turbulent kinetic energy due to average velocity gradient |
| $G_b$ | Turbulent kinetic energy due to buoyancy |
| $Y_M$ | Fluctuating expansion on the total dissipation rate in compressible turbulence |
| $C_2$ | Constant |
| $C_{1\varepsilon}$ | Constant |
| $\zeta_\alpha$ | Constant |
| $\zeta_\varepsilon$ | Constant |
| M | Third body efficiencies |
| $C_{ex}^*$ | Actual characteristic velocity liquid oxygen |
| $C_{th}^*$ | Theoretical characteristic velocity |
| $P_c$ | Gas generator pressure |
| $A_t$ | Nozzle throat area |
| $\dot{m}$ | Propellant mass flow |

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
