# Peer review of "Effects of Local Mixing Ratios and Mass Flow Rates on Combustion Performance of the Fuel-Rich LOX (Liquid Oxygen)/kerosene Gas Generator in the ATR (Air Turbo Rocket) Engine"

_aerospace, doi:10.3390/aerospace10060545_

Round 1
Reviewer 1 Report
The authors describes the results of CFD simultations conducted in a combustion chamber of ATR egine fueled with LOX/kerosene in fuel-rich mode. With the aim of studying the impact of local mixing ratio and propellant/oxidizer flow-rates, different conditions were investigated. The results have shown different zones characterized by very different temperature and chemical componentes have been identified. By the analysis of the results, the authors tried to justify the output of the simulations.
The paper is very problematic because of the poor level of writing that hinders the comprehension of the readers. Many typos, spelling and grammar mistakes are prestent and persistent along the paper. This reviewer suggests an extensive english review.
Major concerns were found regarding the motivations of the paper. Both the abstract and the introduction do not clearly focus the attention on the motivations and, then, the aim of the work is consequently not clear.
Specific comments are reported below:
1. This reviewer suggests to rearrange the abstract in a more concise way highlighting the aim of this work.
2. The introduction is too generic and it is far from the aim of the work. Please rewrite.
3. The numerical model has been verified testing GO2/kerosene combustion in a different combustion chamber with respect to the combustion chamber considered in this work (different injectors number, positions,...). Regarding the validation process, the authors claim that the simultaion model has been validated, however any results are reported and described in the paper. Moreover, the results of the simultations are referred to LOX/kerosene while the simultation model has been validated with GO2/kerosene. Please justify.
4. The combustion efficiency has been defined through equation 3 as velocity ratio. Please extend the combustion efficiency definition.
5. Please define the relative error described at line 254.
6. Please avoid repretitions: lines 243-251.
7. Please give a quantitative explaination of "sufficient combustion"/ "most sufficient" (i.e. line 511)
8. Please define the stoichometric mass of oxigen for the kerosene
9. Please improve the clarity when describing figures.
10. Please capitalize the Section titles (line 624)
11. Please be sure that all the figures and legends have the same font.
12. Please improve the caption of all the figures reported in the paper adding the description of each subplot.
13. Please add the symbols and abbreviations section
14. Please improve the conclusion section: it is poor of details and it is not well connected with the previous sections. In order to better understand the aim of this work, Please include some further develpments related to the presented activity as well.
15. Tables 2 and 4 shows differen units for m. Please correct.
16. This reviewer suggest to improve the quality of the reference of this work: in most of these, the doi is not reported. Moreover, many of these are very old and not specifically focused on the topic.
17. Please add the reference of the experiments GO2/kerosene experiment.
18. Please add the reference of the 10 steps Kerosene/O2 combustion mechanism.
Author Response
Dear Editor and Reviewers,
Kindly find attached the reply to the comments and questions posed by the reviewers. We would like to deeply and sincerely thank both reviewers and editors for their valuable input concerning the manuscript (aerospace-2313294). We trust we have addressed all comments as expected. The most relevant changes to the original manuscript have been marked in red in the current version.
Yours sincerely,
Bin Hu
Response to Reviewer 1 Comments
Point 1: The paper is very problematic because of the poor level of writing that hinders the comprehension of the readers. Many typos, spelling and grammar mistakes are prestent and persistent along the paper. This reviewer suggests an extensive english review.
Response 1: The typos, spelling and grammar mistakes rewritten in revised version and marked in red.
Point 2: Major concerns were found regarding the motivations of the paper. Both the abstract and the introduction do not clearly focus the attention on the motivations and, then, the aim of the work is consequently not clear.
Response 2: Both the abstract and the introduction were rewritten in revised version and marked in red. This paper aim to study the effects of local mixing ratios and local mass flow rates on the combustion performance, which provides the technical support for the optimization design of the LOX/kerosene fuel-rich gas generator in ATR engine.
Point 3: This reviewer suggests to rearrange the abstract in a more concise way highlighting the aim of this work.
Response 3: The abstract rewritten in revised version and marked in red. Lines 12-26 in abstract are modified as " With the increase of the mixing ratio in the inner-ring injectors (meanwhile, the mixing ratio in the middle-ring injectors is decreased), the oxygen concentration area near the axis zone and 50% radius zone of the gas generator is increased. In the flow direction section, the variation trend of local temperature is opposite to oxygen concentration. With the increase of local oxygen concen-tration the local temperature decrease, With the decrease of local oxygen concentration the local temperature increase. The temperature distribution of the gas generator cross-section presents a circular pattern. When the mixing ratio (or mass flow rates) of the unit injector is completely equal, the temperature distribution is the most uniform. With the increase of the mixing ratio in the inner-ring injectors (meanwhile, the mixing ratio in the middle-ring injectors is decreased), the combustion efficiency increases first and then decreases, When the mixing ratio of Km-1=0.7, Km-2=2.79, the combustion efficiency reaching the maximum of 89.10%. With the increase of the flow rate in the inner-ring injectors (meanwhile, the flow rate in the middle-ring injectors is de-creased), the combustion efficiency increases first and then decreases When the mass flow rate of m-1= m-2=0.1kg/s, the combustion efficiency reaching the maximum of 86.13%."
Point 4: The introduction is too generic and it is far from the aim of the work. Please rewrite.
Response 4: The introduction rewritten in revised version and marked in red.
Point 5: The numerical model has been verified testing GO2/kerosene combustion in a different combustion chamber with respect to the combustion chamber considered in this work (different injectors number, positions,...). Regarding the validation process, the authors claim that the simultaion model has been validated, however any results are reported and described in the paper. Moreover, the results of the simultations are referred to LOX/kerosene while the simultation model has been validated with GO2/kerosene. Please justify.
Response 5: (1) Because the gasification temperature of liquid oxygen is very low at 90 k, it is quickly gasified at the working temperature of the gas generator and becomes gas oxygen. (2) The shape and characteristic length of the GO2/kerosene torch igniter are close to those of the LOX/kerosene gas generator, both the injectors type is internal-mixing bipropellant swirl injectors, and both belong to the same type of combustion chamber. (3) Both the GO2/kerosene torch igniter and the LOX/kerosene gas generator are fuel-rich combustions, and their combustion mechanisms are the same. The mixing ratio of the GO2/kerosene torch igniter is 0.454-0.656, and the mixing ratio of the inner-ring of the LOX/kerosene gas generator in the No. 1 and No. 2 is 0.43 and 0.70, which is very close.
Point 6: The combustion efficiency has been defined through equation 3 as velocity ratio. Please extend the combustion efficiency definition.
Response 6: The combustion efficiency definition has been extended at Lines 258-265 in the revised version and marked in red. Lines 258-265 are modified as:" In the actual combustion chamber, when the nozzle throat area and the propellant mixing ratio is determined, due to the limitation of the design level, there are always a series of losses in the process of propellant conversion from chemical energy to thermal energy. In other words, the mass flow rate of the same propellant cannot obtain the total pressure of the combustion chamber. In order to evaluate the loss degree in the energy conversion process of the combustion chamber, the concept of combustion efficiency is introduced. Combustion efficiency refers to the ratio of actual characteristic velocity to theoretical characteristic velocity. "
Point 7: Please define the relative error described at line 254.
Response 7: The relative error is defined at Lines 279-280 in the revised version and marked in red. Relative error = ( numerical efficiency-experimental efficiency ) / experimental efficiency × 100 %
Point 8: Please avoid repretitions: lines 243-251.
Response 8: This part has been deleted in the revised version, in Figure 7 above.
Point 9: Please give a quantitative explaination of "sufficient combustion"/ "most sufficient" (i.e. line 511)
Response 9: In this paper defines "sufficient combustion" as Full combustion is more fully converted from chemical energy to thermal energy, and the combustion temperature in this area is higher. Meanwhile, define "sufficient combustion" as the combustion temperature in this area is the highest.
Point 10: Please define the stoichometric mass of oxigen for the kerosene
Response 10: In this paper define the stoichometric mass of oxygen as mass weight average at the different cross-sections.
Point 11: Please improve the clarity when describing figures.
Response 11: The describing figures were rewritten in the revised version of section 3 and marked in red.
Point 12: Please capitalize the Section titles (line 624).
Response 12: The Section titles (line 624) were rewritten in the revised version of lines 649(section “Conclusions”) and marked in red.
Point 13: Please be sure that all the figures and legends have the same font.
Response 13: This part has been modified in the revised version of section 3.2 and marked in red.
Point 14: Please improve the caption of all the figures reported in the paper adding the description of each subplot.
Response 14: This part has been modified in the revised version of section 3 and marked in red.
Point 15: Please add the symbols and abbreviations section.
Response 15: This part has been rewritten in the revised version in lines 679-690(after conclusions)and marked in red.
Point 16: Please improve the conclusion section: it is poor of details and it is not well connected with the previous sections. In order to better understand the aim of this work, Please include some further develpments related to the presented activity as well.
Response 16: The conclusion Section add some datails and connected with the previous sections, this part were rewritten in the revised version of lines 655-673 and marked in red. The further develpments were rewritten in the revised version of lines 674-678 and marked in red.
Point 17: Tables 2 and 4 shows differen units for m. Please correct.
Response 17: This part has been rewritened in the revised version in Table.4 and marked in red.
Point 18: This reviewer suggest to improve the quality of the reference of this work: in most of these, the doi is not reported. Moreover, many of these are very old and not specifically focused on the topic.
Response 18: The references have been modified to add and replace some references in the revised version in lines 56-81 and marked in red.
Point 19: Please add the reference of the experiments GO2/kerosene experiment.
Response 19: This part has been rewritened in the revised version in section 2.4 and marked in red.
Point 20: Please add the reference of the 10 steps Kerosene/O2 combustion mechanism.
Response 20: This part has been rewritened in the revised version in lines199-202 and marked in red.

Reviewer 2 Report
The efficiency of gas combustion in the afterburner of an Air Turbo Rocket (ATR) engine directly affects engine performance. In this regard, it is of great importance to study the mechanism of the reaction of mixing the combustion of gas and air in the afterburner, as well as the regularities in the distribution of local parameters of the flow field.
The paper is dedicated to numerical simulation-based analysis of the effects of local mixing ratios and mass flow rates on combustion performance of fuel-rich LOX/kerosene gas generator in air turbo rocket engine.
However, in comparison with the previous published papers, this paper does not uncover the new phenomena.
It is not obvious from the content of the paper that the obtained simulation results reveal new patterns in comparison with previous studies.
The scientific novelty of the study described in this paper is not obvious.
In the abstract of the paper and in the conclusion, the scientific novelty of this study should be noted.
Almost all references to similar studies are very old and do not reflect the current level of research. It is necessary to provide references to current research in this area:
- V. Fernández-Villacé, G. Paniagua, J. Steelant, Installed performance evaluation of an air turbo-rocket expander engine, Aerospace Science and Technology, Volume 35, 2014, Pages 63-79, https://doi.org/10.1016/j.ast.2014.03.005.
- Shangchun Wang, Yang Liu, Yue Wu, Zheng Ni, Numerical Simulation of Mixing and Combustion Characteristics based on Air Turbine Rocket Engine Rocket Afterburner, Vol. 1 (2022): 5th International Conference on Computer Engineering, Information Science & Application Technology (ICCIA 2022), DOI: https://doi.org/10.54097/hset.v1i.444
- Shangchun Wang, Yang Liu, Yue Wu, Zheng Ni, Working Fluid modeling Based on Air Turbine Rocket Engine and Propellant Comparative Analysis, Highlights in Science, Engineering and Technology: Vol. 1 (2022): 5th International Conference on Computer Engineering, Information Science & Application Technology (ICCIA 2022), https://doi.org/10.54097/hset.v1i.439
- Nan Xiangyi, Liu Yi, Ma Yuan, et al. Thermodynamic process and operating characteristics of air turbo rocket engine[J]. Acta Aerodynamica Sinica, 2022, 40(1): 181-189. DOI: 10.7638/kqdlxxb-2022.0033
It is necessary to clarify what software was used for numerical simulation, or was developed independently by the authors of the paper.
The abbreviation “ATR” should be decoded in the title of the paper and in the Abstract.
Authors must include a list of Abbreviations in the paper.
The distributions of parameters of the combustion products flows obtained using mathematical modeling do not contain new patterns and phenomena that may be useful to researchers and developers.
All references are prepared in violation of MDPI requirements and adjustments are needed to match the MDPI style.
The conclusions do not contain information about the effect of regularities in the distribution of local parameters of the flow field on combustion instability.
In connection with the foregoing, the authors are recommended to more accurately formulate the concept and scientific novelty of their research in the paper.
Thank you!

Author Response
Dear Editor and Reviewers,
Kindly find attached the reply to the comments and questions posed by the reviewers. We would like to deeply and sincerely thank both reviewers and editors for their valuable input concerning the manuscript (aerospace-2313294). We trust we have addressed all comments as expected. The most relevant changes to the original manuscript have been marked in red in the current version.
Yours sincerely,
Bin Hu
Response to Reviewer 2 Comments
Point 1: In comparison with the previous published papers, this paper does not uncover the new phenomena. It is not obvious from the content of the paper that the obtained simulation results reveal new patterns in comparison with previous studies. The scientific novelty of the study described in this paper is not obvious.
Response 1: This paper aims to fuel-rich LOX/kerosene gas generator in ATR(Air Turbo Rocket) engine, analyzes the effects of local mixing ratios and local mass flow rates on the combustion performance under fixed total mixing ratio and total flow rate, The relevant results show that, With the increase of the mixing ratio in the inner-ring injectors (meanwhile, the mixing ratio in the middle-ring in-jectors is decreased), the oxygen concentration area near the axis zone and 50% radius zone of the gas generator is increased, the kerosene concentration area near the axis zone is decreased and near the 50% radius zone is slowly increased. In the flow direction section, the variation trend of local temperature is opposite to oxygen concentration. With the increase of local oxygen concen-tration, the local temperature decrease; With the decrease of local oxygen concentration, the lo-cal temperature in-crease. The temperature distribution of the gas generator cross-section pre-sents a circular pattern. When the mixing ratio (or mass flow rates) of the unit injector is com-pletely equal, the temperature distribution is the most uniform. The mass flow rate of between the inner ring injector and the middle ring injector is larger, the combustion efficiency is lower, which is different from the mixing ratio of the inner ring injector and the middle ring injector. With the increase of the mixing ratio in the inner-ring injectors (meanwhile, the mixing ratio in the middle-ring injectors is decreased), the combustion efficiency increases first and then de-creases, When the mixing ratio of Km-1=0.7, Km-2=2.79, the combustion efficiency reaching the maximum of 89.10%. With the increase of the flow rate in the inner-ring injectors (meanwhile, the flow rate in the middle-ring injectors is decreased), the combustion efficiency increases first and then decreases When the mass flow rate of m-1= m-2=0.1kg/s, the combustion efficiency reaching the maximum of 86.13%.
Previous published papers not analyzes the effects of local mixing ratios and local mass flow rates on the combustion performance in fuel-rich LOX/kerosene gas generator.
Point 2: Almost all references to similar studies are very old and do not reflect the current level of research. It is necessary to provide references to current research in this area.
Response 2: we have been adding current research in this area, This part has been rewritened in the revised version in reference and marked in red.
Point 3: It is necessary to clarify what software was used for numerical simulation, or was developed independently by the authors of the paper.
Response 3: In this paper, we used GAMBIT software for grid meshing, used ansys fluent software for numerical simulation, and used Tecplot software for analysis of picture.
Point 4: The abbreviation “ATR” should be decoded in the title of the paper and in the Abstract.
Response 4: This part has been rewritten in the revised version in the title of the paper and marked in red.
Point 5: Authors must include a list of Abbreviations in the paper.
Response 5: This part has been rewritten in the revised version in lines 679-690 and marked in red.
Point 6: The distributions of parameters of the combustion products flows obtained using mathematical modeling do not contain new patterns and phenomena that may be useful to researchers and developers.
Response 6: The main work of this paper is to obtain the influence of different local mixing ratios and local flow rates on combustion performance ( combustion efficiency ) and the reasons, to obtain the best design scheme, which is very necessary for the detailed design of gas generator. Different from the rocket engine, ATR (Air Turbo Rocket)engine has a wide range of velocities and a high range of airspace, which requires the fuel-rich gas generator to have relatively high performance in a wide working range. This requires detailed design of the flow rate and mixing ratio of each nozzle to ensure relatively high performance in the full envelope.
Point 7: All references are prepared in violation of MDPI requirements and adjustments are needed to match the MDPI style.
Response 7: This part has been rewritened in the revised version in reference and marked in red.
Point 8: The conclusions do not contain information about the effect of regularities in the distribution of local parameters of the flow field on combustion instability.
Response 8: The main purpose of this paper is to analyze the effects of different unit nozzle flow rates and mixing ratios on the flow field and combustion performance. The study of combustion instability (combustion oscillation) is not designed, and the study of combustion instability will be considered in future research.
Point 9: In connection with the foregoing, the authors are recommended to more accurately formulate the concept and scientific novelty of their research in the paper.
Response 9: This article is aimed at the ATR engine, which works in a wide range of velocities and a high range of airspace, then the gas generator also needs to maintain efficient combustion in a wide range of working ranges, therefore the unit nozzle mass flow rate and mixing ratio have a great influence on the combustion flow field and combustion performance. To ensure that the gas generator has high combustion performance, it is necessary to finely design the mass flow rate and mixing ratio of each nozzle, but there is no public report in this area.

Round 2
Reviewer 1 Report
Despite a huge work made by the authors, the paper is still problematic. Very often spelling mistakes and typos are present and sometimes hinder the comprehension of the text.
Major concerns are referred to the temperature results shown in the paper. The authors said that the wall temperature and kerosene cooling effects are neglected by the model. However, several comments has been referred to temperature near the walls. Moreover, the combution efficiency and the air/fuel ratio distrbution in the igniter might be affected by such phenomena. This reviwer suggests to clarify your assumption and better explain why you can use temperature results to justify the results.
The second major concers of this reviewer are related to the methodology used to validate the simulation model. The authors said that the combustion model has been validated by using experimental data obtained testing a different torch igniter. Since the model neglect the wall temperature and kerosene cooling effects how the results can be compared with different igniter? Why the authors can consider validated the combustion methodology neglecting such effects?
Below are reported typos and spelling mistakes:
lines: 18,25,60,63(x2),121,122,127,130,134,140,142,143,172(Table.2),179,208,261,275,297,299,305,306(x2),307,328,329(x2),336,350,378,379,386,389,390,394,424,433,453,462,500,528,536(x2),558,559,584,585,617.
Further comments are listed below:
1. The abstract is still too long and confused. It seems a list of results. Please rewrite in a more appropriate way.
2. In order to improve the clarity of the text, please rewrite the following sentences: lines(from-to) 49-50, 137-142, 305-309, 328-330, 362-366,423, 433-440,585-586, 634.
3. Please rewrite the introduction highlighting the limits of the mentioned works. Such details may increase the consistency of the presented work.
4. Please avoid chinese characters in line 73
5. Please define GO2 in the text in line 148
6. Please clarify which are the propellants you are referring to in line 151.
7. To improve the clarity, please mention Figure 2c in line 155.
8. Which species are talinkg about in line 164?
9. Please add some quantitative values about the temperature limit of the wall line 170.
10. Please add references for the SIMPLE alghoritm model line 184-194-
11. Please add the equation number in line 187.
12. Since you neglect some temperature effects (kerosene cooling effect and t wall), please add extensive explaination about the temperature results and why and how the authors can believe in the simulation results.
13. In Figure 7, please add tips idetifying the tested conditions.
14. How the combustion model can be reliable neglecting the kerosene cooling effect and the t wall? such phenomena might affect the combustion efficiency.
15. Please review the fonts in Figure 10 to be consistent with others .
16. Please define and add a numerical reference value for the stoichiometric ratio.
Author Response
Dear Editor and Reviewers,
Kindly find attached the reply to the comments and questions posed by the reviewers. We would like to deeply and sincerely thank both reviewers and editors for their valuable input concerning the manuscript (aerospace-2313294). We trust we have addressed all comments as expected. The most relevant changes to the original manuscript have been marked in blue in the current version.
Yours sincerely,
Bin Hu
Response to Reviewer 1 Comments
Point 1: Major concerns are referred to the temperature results shown in the paper. The authors said that the wall temperature and kerosene cooling effects are neglected by the model. However, several comments has been referred to temperature near the walls. Moreover, the combution efficiency and the air/fuel ratio distrbution in the igniter might be affected by such phenomena. This reviwer suggests to clarify your assumption and better explain why you can use temperature results to justify the results.
Response 1: In this paper, the wall temperature and kerosene cooling effect are not ignored, but under the condition that the total mass flow and total mixing ratio are fixed, the mass flow rate of outer-ring cooling kerosene is also fixed, so as to ensure that the inner wall of gas generator under various working conditions is not damaged by ablation. In this paper, the effect of the mass flow change of outer ring cooling kerosene on the wall temperature is not explored.
The purpose of the torch igniter experimental and simulation comparison is to verify the accuracy of the oxygen/kerosene chemical reaction mechanism and numerical simulation method under fuel-rich conditions. The oxygen/kerosene in the torch igniter is mixed and combusted, and the oxygen/kerosene in the gas generator is also mixed and combusted. The outer-ring cooling kerosene contact with the inner-ring and middle-ring propellent will undergo a combustion reaction, but the mixing ratio of the outer-ring cooling kerosene is zero, which cannot be fully involved in combustion. The kerosene near the wall forms a kerosene protective layer to protect the inner wall of the gas generator.
In the torch igniter experimental and simulation, the maximum relative error (Relative error = (numerical efficiency-experimental efficiency)/experimental efficiency×100 %) is 1.41%, the average error is 0.61%, and the error is within the acceptable range. Therefore, the computational method in this paper is feasible.
Point 2: The second major concers of this reviewer are related to the methodology used to validate the simulation model. The authors said that the combustion model has been validated by using experimental data obtained testing a different torch igniter. Since the model neglect the wall temperature and kerosene cooling effects how the results can be compared with different igniter? Why the authors can consider validated the combustion methodology neglecting such effects?
Response 2: For the torch igniter, due to the short ignition time, low mixing ratio, and low combustion temperature, the torch igniter adopts the heat sink cooling method, and does not set cooling kerosene to cool the wall. The purpose of the comparison between the experimental and simulation of the torch igniter is to verify the accuracy of the oxygen-kerosene chemical reaction mechanism and the numerical simulation method under the fuel-rich combustion reaction. The oxygen-kerosene in the torch igniter is mixed and occurs combustion reaction. The parameters such as combustion temperature and pressure are also different under different mixing ratios. In the torch igniter experimental and simulation, the maximum relative error (Relative error = (numerical efficiency-experimental efficiency)/experimental efficiency×100 %) is 1.41%, the average error is 0.61%, and the error is within the acceptable range. Therefore, the computational method in this paper is feasible.
Point 3: Below are reported typos and spelling mistakes: lines:18,25,60,63(x2),121,122,127,130,134,140,142,143,172(Table.2),179,208,261,275,297,299,305,306(x2),307,328,329(x2),336,350,378,379,386,389,390,394,424,433,453,462,500,528,536(x2),558,559,584,585,617.
Response 3: The typos and spelling mistakes rewrite in revised version and marked in blue.
Point 4: The abstract is still too long and confused. It seems a list of results. Please rewrite in a more appropriate way.
Response 4: The abstract has been rewrite in revised version in lines 17-25 and marked in blue.
Point 5: In order to improve the clarity of the text, please rewrite the following sentences: lines(from-to) 49-50, 137-142, 305-309, 328-330, 362-366,423, 433-440,585-586, 634.
Response 5: This part has been rewrite in the revised version and marked in blue, rewrite in lines 51-53, 146-152, 319-323, 342-344, 378-382, 439-440, 449-455, 600-604, 653-655,respectively.
Point 6: Please rewrite the introduction highlighting the limits of the mentioned works. Such details may increase the consistency of the presented work.
Response 6: The abstract has been rewrite in revised version in introduction and marked in blue.
Point 7: Please avoid chinese characters in line 73.
Response 7: This part has been deleted in the revised version in lines 76 and marked in blue.
Point 8: Please define GO2 in the text in line 148.
Response 8: This part has been defined in Lines 158 in the revised version and marked in blue.
Point 9: Please clarify which are the propellants you are referring to in line 151.
Response 9: This part has been rewrite in the revised version in lines 161-162 and marked in blue, modified as "The propellant of the gas generator is LOX / kerosene, the propellant of the torch ignitor is GO2/kerosene.”
Point 10: To improve the clarity, please mention Figure 2c in line 155.
Response 10: This part has been rewrite in the revised version in lines 165-166 and marked in blue, modified as "Figure 2(c) is the Injector arrangement of gas generator.”
Point 11: Which species are talinkg about in line 164?
Response 11: This part has been rewrite in the revised version in lines 175-176 and marked in blue, modified as "The total mass flow rate and mixing ratio of LOX and kerosene in the gas generator at the design point are 2.26kg/s and 1, respectively. ”
Point 12: Please add some quantitative values about the temperature limit of the wall line 170.
Response 12: This part has been rewrite in the revised version in lines 181-182 and marked in blue, modified as “The inner wall temperature does not exceed 1200k.”
Point 13: Please add references for the SIMPLE alghoritm model line 184-194-
Response 13: The references[27],[28] for the SIMPLE alghoritm model has been added in the revised version in lines 195 and 206, and marked in blue.
Point 14: Please add the equation number in line 187.
Response 14: This part has been modified in the revised version in line 198 and marked in blue.
Point 15: Since you neglect some temperature effects (kerosene cooling effect and t wall), please add extensive explaination about the temperature results and why and how the authors can believe in the simulation results.
Response 15: In this paper, the wall temperature and kerosene cooling effect are not ignored, but under the condition that the total mass flow and total mixing ratio are fixed, the mass flow rate of outer-ring cooling kerosene is also fixed, so as to ensure that the inner wall of gas generator under various working conditions is not damaged by ablation. In this paper, the effect of the mass flow change of outer ring cooling kerosene on the wall temperature is not explored.
The purpose of the torch igniter experimental and simulation comparison is to verify the accuracy of the oxygen/kerosene chemical reaction mechanism and numerical simulation method under fuel-rich conditions. The oxygen/kerosene in the torch igniter is mixed and combusted, and the oxygen/kerosene in the gas generator is also mixed and combusted. The outer-ring cooling kerosene contact with the inner-ring and middle-ring propellent will undergo a combustion reaction, but the mixing ratio of the outer-ring cooling kerosene is zero, which cannot be fully involved in combustion. The kerosene near the wall forms a kerosene protective layer to protect the inner wall of the gas generator.
In the torch igniter experimental and simulation, the maximum relative error (Relative error = (numerical efficiency-experimental efficiency)/experimental efficiency×100 %) is 1.41%, the average error is 0.61%, and the error is within the acceptable range. Therefore, the computational method in this paper is feasible.
Point 16: In Figure 7, please add tips idetifying the tested conditions.
Response 16: This part has been added in the revised version in Figure 7 in line 282 and marked in blue.In Figure 7, the abscissa and ordinate are numerical simulation results and experimental results of combustion efficiency under four conditions, respectively. When the point in the figure is below the reference slash, the simulated value is higher than the experimental value. When the point in the figure is above the reference slash, the experimental value is higher than the simulated value. The closer the point in the figure is to the reference line, the closer the test value is to the simulated value.
Point 17: How the combustion model can be reliable neglecting the kerosene cooling effect and the t wall? such phenomena might affect the combustion efficiency.
Response 17: The purpose of the torch igniter experimental and simulation comparison is to verify the accuracy of the oxygen/kerosene chemical reaction mechanism and numerical simulation method under fuel-rich conditions. The oxygen/kerosene in the torch igniter is mixed and combusted, and the oxygen/kerosene in the gas generator is also mixed and combusted. The outer-ring cooling kerosene contact with the inner-ring and middle-ring propellent will undergo a combustion reaction, but the mixing ratio of the outer-ring cooling kerosene is zero, which cannot be fully involved in combustion. The kerosene near the wall forms a kerosene protective layer to protect the inner wall of the gas generator.
In the torch igniter experimental and simulation, the maximum relative error (Relative error = (numerical efficiency-experimental efficiency)/experimental efficiency×100 %) is 1.41%, the average error is 0.61%, and the error is within the acceptable range. Therefore, the computational method in this paper is feasible.
Point 18: Please review the fonts in Figure 10 to be consistent with others.
Response 18: The fonts in Figure 10 has been modified in the revised version in line 366 and marked in blue.
Point 19: Please define and add a numerical reference value for the stoichiometric ratio.
Response 19: This part has been add define “stoichiometric ratio” as “When fuel and air are completely burned, the ratio of fuel quality to air quality.” in lines 317 and marked in blue.

Reviewer 2 Report
The revised manuscript looks much better.
The abbreviations “LOX” and “ATR” should be decoded in the Abstract.
References to the software used should be given in the text of the manuscript: GAMBIT software and Ansys Fluent software and Tecplot software.
It is also necessary to provide relevant references on software used in the List of References.
The list of references must include all symbols used in formulas. Accordingly, the list of symbols given in the text should be transferred to the list of references.
And again, all references are prepared in violation of MDPI requirements and adjustments are needed to match the MDPI style.
The List of References should be corrected.
Thank you!
Author Response
Dear Editor and Reviewers,
Kindly find attached the reply to the comments and questions posed by the reviewers. We would like to deeply and sincerely thank both reviewers and editors for their valuable input concerning the manuscript (aerospace-2313294). We trust we have addressed all comments as expected. The most relevant changes to the original manuscript have been marked in blue in the current version.
Yours sincerely,
Bin Hu
Response to Reviewer 2 Comments
Point 1: The abbreviations “LOX” and “ATR” should be decoded in the Abstract.
Response 1: The “LOX” decoded in (Liquid Oxygen) and “ATR” decode in (Air Turbo Rocket), This part has been rewritten in the revised version in line 11 and marked in blue.
Point 2: References to the software used should be given in the text of the manuscript: GAMBIT software and Ansys Fluent software and Tecplot software.
Response 2: This part has been rewritten in the revised version of 2.4 - 2.5 and marked in blue. Gambit software references in line 227, Ansys Fluent software references in line 237, and Tecplot360 software references in line 238.
Point 3: It is also necessary to provide relevant references on software used in the List of References.
Response 3: This part has been added in References [30-32] in lines 799-802 and marked in blue.
Point 4: The list of references must include all symbols used in formulas. Accordingly, the list of symbols given in the text should be transferred to the list of references.
Response 4: All symbols used in formulas has been added in Nomenclature. This part has been rewritten in the revised version in lines 710-726 and marked in blue.
Point 5: And again, all references are prepared in violation of MDPI requirements and adjustments are needed to match the MDPI style.
Response 5: This part has been rewritten in the revised version in lines 738-802 and marked in blue.
Point 6: The List of References should be corrected.
Response 6: we have been corrected the list of References, this part has been rewritten in the revised version in lines 738-802 and marked in blue.

Round 3
Reviewer 1 Report
This reviewer appreciated the huge work made by the authors discussing all the points mentioned in the previous version. however, before publishing the paper I strongly recommend to perform an extensive english review by a native speaker.
regards,
GS
Author Response
Dear Editor and Reviewers,
Kindly find attached the reply to the comments and questions posed by the reviewers. We would like to deeply and sincerely thank both reviewers and editors for their valuable input concerning the manuscript (aerospace-2313294). We trust we have addressed all comments as expected. The most relevant changes to the original manuscript have been marked in green in the current version.
Yours sincerely,
Bin Hu
Response to Reviewer 1 Comments
Point 1: This reviewer appreciated the huge work made by the authors discussing all the points mentioned in the previous version. however, before publishing the paper I strongly recommend to perform an extensive english review by a native speaker.
Response 1: This paper has been perform an extensive english review by a native speake in the revised versionand marked in green.

Reviewer 2 Report
The revised manuscript looks much better.
However, again, the symbols given in the text (lines 199-204, 279, 280, 281) should be excluded from the text and transferred to the Nomenclature section.
Thank you!
Author Response
Dear Editor and Reviewers,
Kindly find attached the reply to the comments and questions posed by the reviewers. We would like to deeply and sincerely thank both reviewers and editors for their valuable input concerning the manuscript (aerospace-2313294). We trust we have addressed all comments as expected. The most relevant changes to the original manuscript have been marked in green in the current version.
Yours sincerely,
Bin Hu
Response to Reviewer 2 Comments
Point 1: The symbols given in the text (lines 199-204, 279, 280, 281) should be excluded from the text and transferred to the Nomenclature section.
Response 1: The symbols given in the text (lines 199-204, 279, 280, 281) has been excluded from the text and transferred to the Nomenclature section in the revised version in line 701-710 and marked in green.
